# A simple method for developing lysine targeted covalent protein reagents

Ronen Gabizon[1,4], Barr Tivon[1,4], Rambabu N. Reddi[1],
Maxime C. M. van den Oetelaar[2], Hadar Amartely[3], Peter J. Cossar [2],
Christian Ottmann [2] & Nir London [1] ✉

Peptide-based covalent probes can target shallow protein surfaces not typically addressable using small molecules, yet there is a need for versatile approaches to convert native peptide sequences into covalent binders that can target a broad range of residues. Here we report protein-based thio-methacrylate esters—electrophiles that can be installed easily on unprotected peptides and proteins via cysteine side chains, and react efficiently and selectively with cysteine and lysine side chains on the target. Methacrylate phospho-peptides derived from 14-3-3-binding proteins irreversibly label 14-3-3σ via either lysine or cysteine residues, depending on the position of the electrophile. Methacrylate peptides targeting a conserved lysine residue exhibit pan-isoform binding of 14-3-3 proteins both in lysates and in extracellular media. Finally, we apply this approach to develop protein-based covalent binders. A methacrylate-modified variant of the colicin E9 immunity protein irreversibly binds to the E9 DNAse, resulting in significantly higher thermal stability relative to the non-covalent complex. Our approach offers a simple and versatile route to convert peptides and proteins into potent covalent binders.

Covalent tool compounds and chemical probes have been established as a powerful technology with diverse applications in chemical biology. These applications range from inhibitors used for therapeutic applications[1–6] to covalent probes used to study the function and properties of target proteins. Covalent compounds have been developed targeting a large variety of proteins including kinases[1–4,7–10], G-protein coupled receptors[11,12], hydrolases[13–17], have found important uses as probes for proteomics[18–20] and microscopy[11], and are also used in emerging applications such as targeted degradation[21–27]. The advantages of covalent compounds in chemical biology stem from several aspects. The irreversible binding to the target achieves prolonged potent inhibition with short systemic exposure[4,28,29]. Covalent binding to the target facilitates downstream processes involving denaturation and proteolysis of the target without loss of the bound probe, making them especially useful in proteomics. Lastly, covalent

binders frequently show enhanced selectivity by targeting non-conserved nucleophilic residues. This is exemplified by the recently approved Sotorasib[5] and Adagrasib[6], which target Kras[G12C].

Despite the surge in research into covalent compounds, targets such as transcription factors and protein–protein interaction interfaces are difficult to target with small molecules due to their broad and shallow binding surfaces[30,31]. The use of peptide or peptidomimetics has emerged as a powerful approach to address these issues. Peptide binders can cover a large surface area, can bind protein targets with high affinity and can frequently be derived from known protein–protein interactions[32–38]. Potent covalent peptide binders have been developed for targets such as the bacterial divisome[39], E3 ubiquitin ligases[40], the anti-apoptotic protein BFL-1[41], and others[42–47].

Due to the size of peptides and their conformational flexibility, computational modeling can aid in the design and placement of the

[1]Department of Chemical and Structural Biology, The Weizmann Institute of Science, Rehovot 7610001, Israel. [2]Laboratory of Chemical Biology, Department of Biomedical Engineering, Institute for Complex Molecular Systems, Eindhoven University of Technology, P.O. Box 513, 5600MB Eindhoven, The Netherlands. [3]Wolfson Centre for Applied Structural Biology, The Alexander Silberman Institute of Life Sciences, The Hebrew University of Jerusalem, Jerusalem 9190401, Israel. [4]These authors contributed equally: Ronen Gabizon, Barr Tivon. ✉e-mail: nir.london@weizmann.ac.il

electrophile. Computational modeling has been used extensively to model and design peptide and peptidomimetic binders for proteins[48–54]. We developed CovPepDock[55], a Rosetta-based framework for modeling covalent protein-peptide interactions and for the design and virtual screening of potential covalent peptide binders. Using CovPepDock we designed acrylamide and chloroacetamide peptide binders that target the non-conserved Cys38 residue in the σ isoform of the 14-3-3 family. The peptides displayed highly potent and selective detection of 14-3-3σ in cell lysates, a difficult task for non-covalent binders due to the high sequence homology within the family.

While aiding the design of selective binders, the low abundance of cysteine also presents a problem and excludes many potential targets. In the case of 14-3-3 proteins, the different isoforms are highly similar in sequence and there is some degree of functional redundancy in their activity[56,57]. Therefore it is also desirable to develop probes that label all 14-3-3 proteins in the family. A surge in the development of novel warheads for small molecules has expanded the targetable scope of amino acids to include lysine[10,20,58], tyrosine[59], acidic residues[60], histidine[61], and others[62]. These chemistries have greatly expanded the spectrum of protein targets. However, the synthetic installation of a reactive group on a peptide or full protein is not trivial. The large number of nucleophilic amino acids on a protein or the relatively harsh deprotection conditions required for solid phase peptide synthesis complicates electrophile installation. Various approaches have emerged recently to functionalize native sequences and prepare protein-based covalent binders. These include genetic code expansion to incorporate reactive groups into protein sequences via unnatural amino acids[63–68], approaches utilizing enzymatic activation of proteins[69,70] and site-selective chemical approaches[71–76]. However, these techniques remain technically challenging, and there remains a need for simple approaches to expand the scope of targetable residues with high selectivity.

This work describes an approach to the development of covalent protein reagents based on thioether-modified methacrylate electrophiles. These mild electrophiles can react with both lysine and cysteine side chains via Michael addition. The electrophile is installed by direct, selective modification of a cysteine side chain, enabling synthesis of binders from unprotected peptides and even recombinant proteins (Fig. 1). We applied this approach to prepare pan-14-3-3 covalent binders that label a conserved lysine in the peptide binding pocket[77]. The peptides displayed efficient and selective binding of 14-3-3 proteins[56,78] in cell lysates as well as secreted 14-3-3 proteins, extracellularly. We then expanded this approach to proteins and prepared a modified mutant of the E9 colicin immunity protein[79,80], which covalently bound the E9 Nuclease. The covalent complex displayed dramatically higher thermal stability than the non-covalent complex. This tool offers a versatile approach to the design and preparation of potent covalent binders for diverse targets.

## Results

Two key features were critical to our design criteria for robust generation of covalent protein reagents. First, electrophile installation should be performed on unprotected peptides or proteins. Second, the electrophile should react with both thiols and primary amines. As such we selected substituted methacrylamides or methacrylates that react with both cysteine[81], and lysine residues via aza-Michael addition[82]. Further, we selected ethyl 2-(bromomethyl)acrylate (Fig. 2c) for its selective reactivity with cysteine residues via its α-bromo-methylene functional group. Cysteine residues represent an attractive anchoring residue due to their low proteomic abundance and their high reactivity at physiological pH, enabling rapid and selective modification on unprotected peptides and proteins.

### Design and synthesis of methacrylate peptide binders against 14-3-3 proteins

Our primary objective was to design binders that would target conserved lysines in the phoshopeptide binding groove of 14-3-3σ: lysines 49 and 122 (Fig. 2a, b). Specifically, Lys122 has been previously shown to react with aldehydes to form a reversible covalent imine bond, with high selectivity over other lysine residues, which is attributed to a lower pKa of the Lys122 side-chain[83–85].

We first used CovPepDock to design a series of methacrylate-based peptides based on the non-covalent complex between 14-3-3σ and YAP1 phosphopeptide (PDB: 3MHR), as we have used for designing our previously reported Cys38 binding peptides. We identified the residues in the YAP1 peptide that are within Cα-Cα distance <14 Å from the target lysine and mutated each of these residues to the methacrylate modified side-chain; this included residues 126–131 for targeting Lys49, and residues 126–133 for targeting Lys122. We selected four peptides that were predicted to bind either Lys49 or Lys122 with high score and low Root Mean Square Deviation (RMSD) to the original peptide binding mode (i.e., predicted to bind well and maintain the binding pose of the original peptide). To further expand the scope of series, we designed a second set of peptides based on other known peptide binders of 14-3-3σ. We selected four such structures, with peptides derived from Raf1 (PDB: 3IQU and 4IEA), TASK-3 (PDB: 3P1N), and SNAI1 (PDB: 4QLI). For these peptides we focused on Lys122, which is more reactive towards electrophiles[83–85]. We used CovPepDock to model Lys122-targeting peptides based on each of these structures, and selected seven additional peptides with high scores and low RMSD from this set for synthesis and testing (Table 1; Supplementary Table 1).

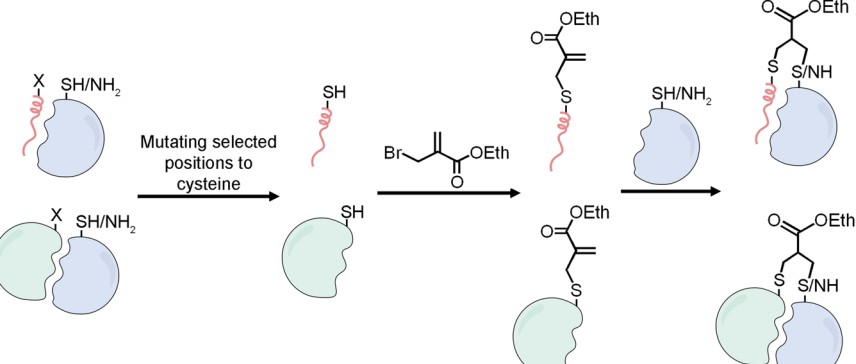

**Fig. 1 | Scheme for generating thioether-based methacrylate covalent peptide or protein binders.** A cysteine residue can be introduced into peptides (pink ribbon) or recombinant proteins (light green), which are then modified using ethyl 2- (bromomethyl)acrylate. The resulting electrophile reacts with lysine or cysteine side chains proximal to the binding site on the receptor (light blue), to yield covalent adducts.

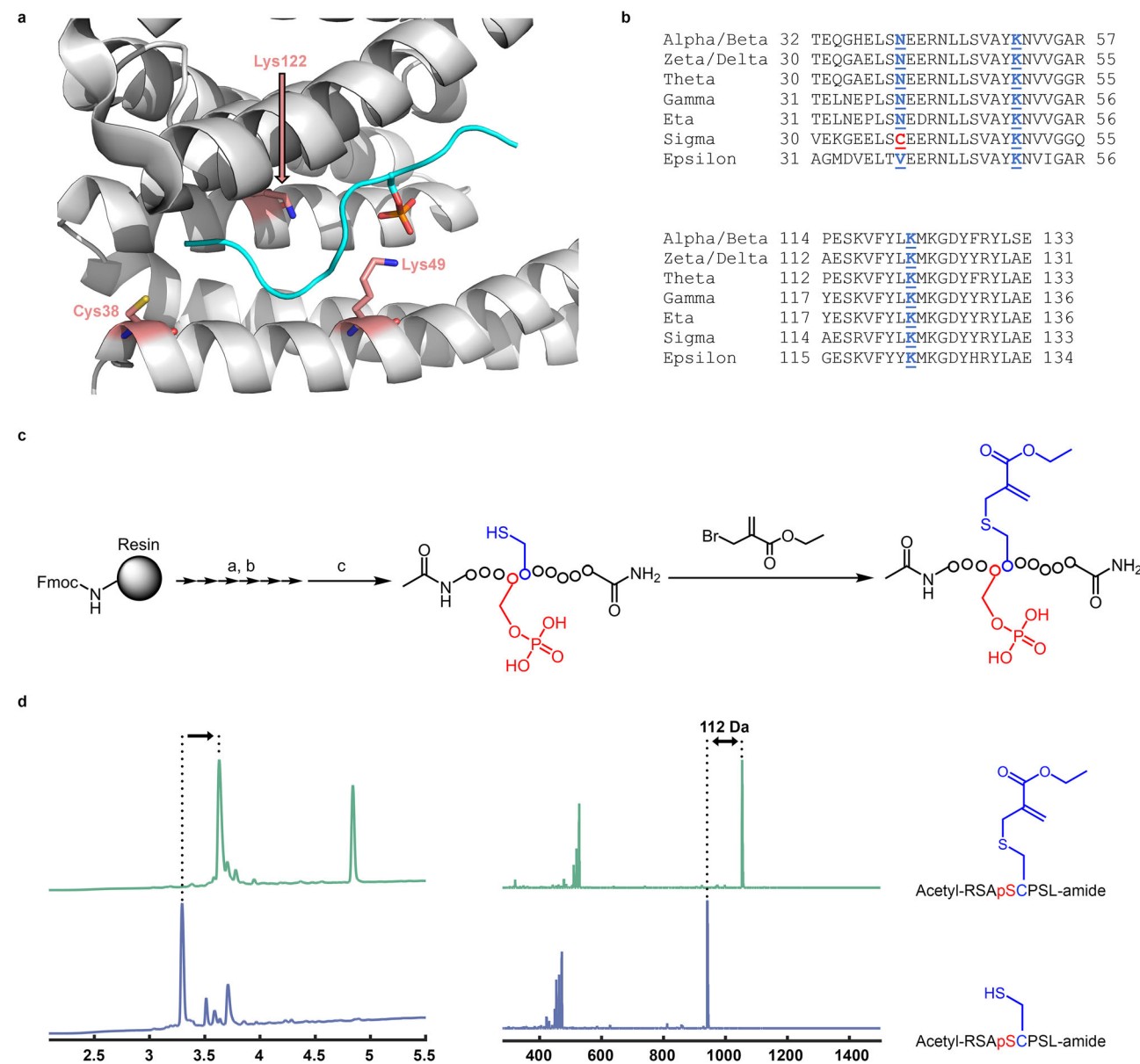

**Fig. 2 | Generating peptide covalent reagents for 14-3-3 proteins. a** Structure of the complex of 14-3-3σ (white) with phosphorylated peptide from YAP (cyan; PDB: 3MHR). The non-conserved cysteine 38 and the conserved lysines 49 and 122 are highlighted. **b** Sequence alignment of 14-3-3 isoforms. Cys38 in 14-3-3σ is unique while lysines 49 and 122 are fully conserved. **c** Scheme for synthesis of electrophilic peptides. (**a**) 20% piperidine in DMF, 3 × 4 min; (**b**) 4 equ. Fmoc-AA-OH/HATU/HOAT, 8 equ. DIPEA, 30 min RT, (**c**) 94% TFA, 2.5% water, 2.5% TIPS, 1% DODT, 3 h. **d** HPLC chromatograms and MS spectra of the crude peptide **3** (dark blue) and the crude peptide after reaction with 2-(bromomethyl)acrylate (green). The second peak is excess 2-(bromomethyl)acrylate.

To prepare peptide methacrylate adducts, we synthesized the peptides using standard Solid Phase Peptide Synthesis (SPPS) procedures and N-terminally acetylated the peptides (Fig. 2c). After cleavage from the resin, the crude peptides were reacted with 3 equivalents of 2-(bromomethyl)acrylate in a buffer devoid of amines or thiols to avoid possible side reactions, resulting in efficient conversion of the peptide to the methacrylate within 1–2 h at room temperature (Fig. 2d). Analytical data for the purified peptides is given in the Source data file.

We incubated the peptides at 200 μM with 2 μM 14-3-3σ at 4 °C overnight and monitored the binding using intact protein LC/MS (Fig. 3a). Peptide **12**, which contains a chloroacetamide warhead that reacts with 14-3-3σ via Cys38[55], was used as a positive control. Significant covalent labeling of 14-3-3σ with the expected adduct mass was observed for the peptides **3** (51%), **8** (35%), and **11** (57%), all of which were predicted to bind Lys122. Peptide **1** also displayed low levels of

labeling (~10%) at the expected adduct mass, as well an unidentified smaller adduct (139 Da less). At this point it was not clear whether the peptides that did not label 14-3-3σ failed to do so due to diminished non-covalent binding affinity or due to suboptimal positioning of the electrophile within the non-covalent complex. Therefore, we exploited the fact that the formation of the covalent bond is slow and takes place over a time scale of hours, and performed a fluorescence polarization binding experiment using a BDP-labeled peptide derived from YAP-1 that binds 14-3-3σ with an affinity of ~100 nM[55]. We added 14-3-3σ at a concentration of 0.25 μM to premixed fluorescent peptide (5 nM) and electrophilic peptides (5 μM or 200 μM) and measured the fluorescence polarization at 27 °C immediately following the mixture (Supplementary Fig. 1). Only peptides 10 and 11 had sufficient affinity to displace the fluorescent binder at 5 μM, while all peptides except peptide 2 displaced the binder at 200 μM, which is the concentration at

## Table 1 | Sequences and structures of the peptides[a]

| Peptide | Source Protein | PDB | Sequence |
| --- | --- | --- | --- |
| 1 | SNAI1 | 4QLI | Ac-SH-(pT)-(mC)-PS-NH$_2$ |
| 2 | SNAI1 | 4QLI | Ac-SH-(pT)-L-(mC)-S-NH$_2$ |
| 3 | Raf1 | 4IEA | Ac-RSA-(pT)-(mC)-PSL-NH$_2$ |
| 4 | Raf1 | 4IEA | Ac-RSA-(pT)-EP-(mC)-L-NH$_2$ |
| 5 | Raf1 | 4IEA | Ac-RSA-(pT)-EPS-(mC)-NH$_2$ |
| 6 | Raf1 | 3IQU | Ac-QRST-(pT)-(mC)-OH |
| 7 | TASK-3 | 3P1N | Ac-KRRK-(pT)-(mC)-NH$_2$ |
| 8 | YAP-1 | 3MHR | Ac-RAH-(pT)-(mC)-PASLQ-NH$_2$ |
| 9 | YAP-1 | 3MHR | Ac-RAH-(pT)-SP-(mC)-SLQ-NH$_2$ |
| 10 | YAP-1 | 3MHR | Ac-RAH-(pT)-SPA-(mC)-LQ-NH$_2$ |
| 11 | YAP-1 | 3MHR | Ac-RAH-(pT)-SPAS-(mC)-Q-NH$_2$ |
| 12 | YAP-1 | 3MHR | Ac-RAH-(pT)-SPASL-X-NH$_2$ |

[a]pS = Phoshoserine; pT = Phosphothreonine; mC = Methacrylate-modified cysteine (Figs. 1 and 2); X = γ-chloroacetamido-diaminobutyric acid[55].

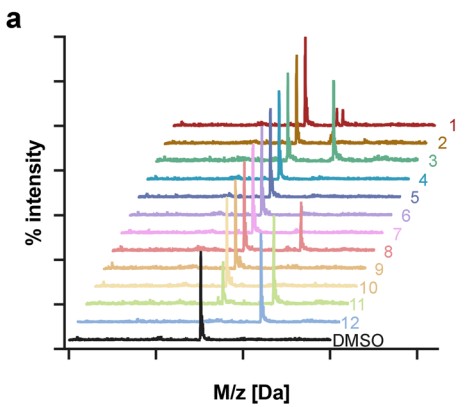

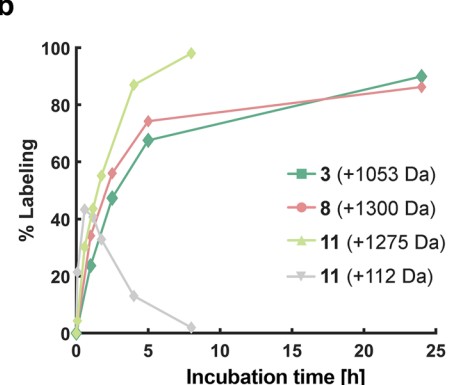

**Fig. 3 | Designed methacrylate peptides bind 14-3-3σ. a** Peptides (200 μM) were incubated with the 14-3-3σ protein (2 μM; overnight; 4 °C) and analyzed using intact protein LC/MS. See Table 1 for peptide structures. **b** Selected peptides (5 μM) were incubated with 14-3-3σ protein (2 μM, room temperature) for different times and analyzed using intact protein LC/MS. **3** and **8** formed only full peptide adducts. **11** also formed a transient methacrylate only adduct which converted to the full adduct over time.

which the initial screen was performed. Therefore, while many of the electrophilic peptides have diminished non-covalent affinity to 14-3-3σ, it is likely that the positioning of the electrophile in the non-covalent complex also plays an important role in covalent bond formation.

We analyzed these peptides further using time course labeling experiments at lower peptide concentrations (5 μM) at 25 °C. Peptides **3** and **8** reached 60% and 80% labeling within 5 h, respectively. Incubation at 37 °C increased the rate of labeling roughly 4-fold (Supplementary Fig. 2). Interestingly, when incubated with **11**, non-labeled 14-3-3σ disappeared rapidly–within 2.5 h less than 5% free 14-3-3σ remained. However, the reaction initially yielded a mixture of full peptide-labeled protein (+1275 Da) and protein modified with only the methacrylate group (+112 Da). The methacrylate-labeled protein was gradually converted to full peptide-labeled protein (Fig. 3b).

We were interested in comparing the intrinsic reactivity and stability of the active peptides with their labeling rates (Supplementary Fig. 3). We therefore tested their stability in buffer and in the presence of lysine and cysteine at room temperature. The peptides showed good stability in buffer–In a time scale of 6 h in buffer, all peptides were more than >90% intact. Over a time scale of days, peptides **8** and **11** formed a new product with the same mass as the original peptide, possibly due to an internal reaction within the peptide. Peptide **3** remained unmodified even after 3 days. Incubation with lysine did not

result in any products other than those observed after prolonged incubation in buffer, indicating the peptides had low intrinsic reactivity towards lysine. The reactivity towards cysteine was much higher –the peptides reacted with cysteine to generate both Michael adducts as well as substitution products in which the thiol peptide is released. Peptides **3**, **8**, and **11** were more than >50% reacted within 2.5 h.

### Methacrylate peptides can target both cysteine and lysine, controlling 14-3-3 isoform selectivity

To elucidate the binding sites of the peptides on 14-3-3σ, we performed trypsin digestion followed by LC/MS/MS. Direct identification and quantification of modified peptides proved challenging due to long peptide chain lengths, fragmentation from multiple directions and weak relative signals. To characterize peptide ligation sites, we switched strategy and measured the relative change in the signal of non-modified peptides on 14-3-3σ relative to a DMSO-treated control. Specifically, we looked at the peptides containing or immediately following residues Cys38, Lys49, and Lys122 (Supplementary Fig. 4A). Peptide **11** specifically reduced the signal of the Cys38-containing peptides, with little effect on the signals for other peptides, indicating specific Cys38 binding. In contrast, peptides containing or following Lys122 were significantly depleted by peptides **3** and **8**, albeit not uniformly, while the signals of Lys49 containing peptides were slightly increased. This result pointed to Lys122 as the likely binding site for

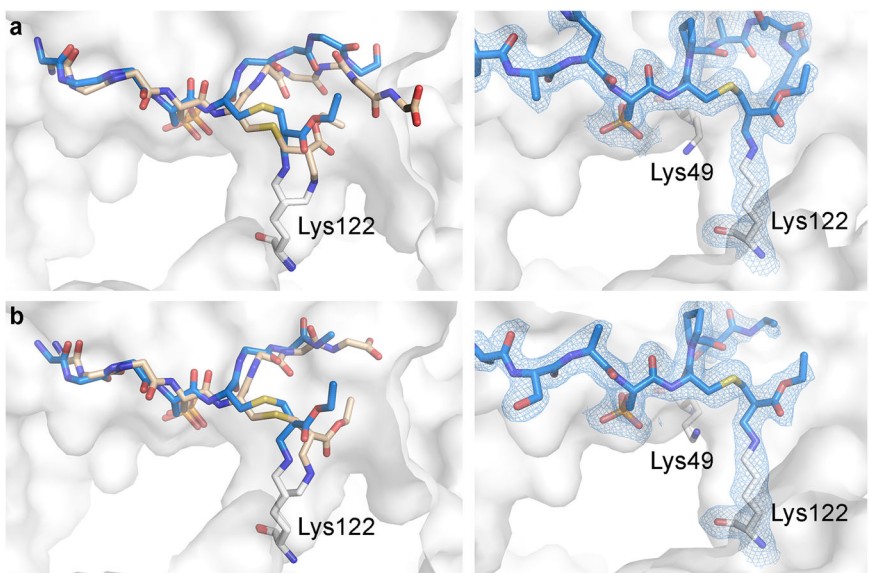

**Fig. 4 | Methacrylate peptides bind conserved 14-3-3 lysine residue. a** Overlay of the CovPepDock prediction (Light brown) and co-crystal structure of peptide **8** (blue sticks) bound to 14-3-3σ (white surface; left), and a close-up view on the methacrylate residue (right). Final 2Fo-Fc electron density (light blue mesh, contoured at 1.0σ) is displayed for peptide **8** (right). Similarly, **b** docking overlay (left) and close-up view (right) of peptide **3**. Final 2Fo-Fc electron density (light blue mesh, contoured at 1.0σ) is displayed for peptide **3** (right).

peptides **3** and **8**. The non-uniform reduction in signal for Lys122-containing peptides may be due to incomplete labeling of 14-3-3σ by **3** and **8**. We also considered the possibility that adducts to lysine are unstable in the presence of reducing agents used before tryptic digest. To test this, we first incubated the peptides with excess TCEP or DTT, which caused rapid release of the peptide from the methacrylate within 2 h (Supplementary Fig. 4B). However, under the same conditions the peptide-protein adducts were stable (Supplementary Fig. 4C). We also incubated the protein with fluorescently labeled methacrylate peptides, followed by denaturation in DTT-containing sample buffer and SDS-PAGE. Here as well, reduction did not affect the intensity of the fluorescent band, indicating that after aza-Michael addition, the protein-peptide adduct is stable (Supplementary Fig. 4D).

To further confirm that Lysine 122 is the target residue for peptides **3** and **8**, we co-crystallized 14-3-3σ with both peptides. The crystal structures (Fig. 4a, b) clearly showed the covalent bond formed between the amine of Lys122 with the methacrylate, with Lys49 remaining unmodified. Comparison of the crystal structure with the prediction from CovPepDock indicated the model correctly predicts the binding pose around the phosphate and the N-terminal part of the peptide, but less so for the C-terminal region. More specifically, compared to the prediction, Lys122 adopts a more relaxed conformation, the C-terminal residues are not as tightly packed in the binding groove, and in peptide **8** (Fig. 4a) the C-terminal glutamine could not be modeled due to insufficient density. We also compared the measured structures of the covalent complexes with the known structures of the non-covalent complexes (Supplementary Fig. 4A, B). For peptide **8**, the C terminal part of the peptide is displaced outwards due to the space occupied by the methacrylate ester moiety. The conformation of the peptide **3** complex is far less affected by covalent binding due to the shorter C terminal part of the peptide. In contrast, both peptides exhibit only a minor effect of covalent binding on the structure of the N terminal region. These results indicate that non-covalent interactions with the C-terminal part of the peptide play a minor role in the binding, in agreement with the behavior observed for chloroacetamide and acrylamide-based covalent peptides we developed previously[55].

Since Lys122 is highly conserved in all 14-3-3 isoforms (in contrast to Cys38 which is unique in 14-3-3σ, Fig. 2b), we incubated peptides **3**

and **8** with other isoforms, together with the Cys38-targeting peptide **12**[55]. While peptide **12** labeled only the sigma isoform, peptides **3** and **8** labeled all isoforms with similar efficiencies (Supplementary Fig. 6), Taken together these results conclusively validate that peptides **3** and **8** specifically bind to Lys122 via aza-Michael addition.

## Methacrylate peptides detect 14-3-3 proteins in lysates and extracellular media with high sensitivity

We have previously shown that electrophilic peptides enable sensitive and selective labeling of 14-3-3σ in cell lysates[55]. Since the methacrylate peptides can react with all 14-3-3 isoforms, we prepared BODIPY-labeled derivatives of peptides **3**, **8**, and **12** and tested if they can function as pan-reactive 14-3-3 probes in cell lysates (Fig. 5a, Supplementary Fig. 7A). Fluorescently labeled **12** selectively labeled 14-3-3σ and generated a single fluorescent band. The methacrylates **3** and **8** formed two main bands, with the bottom band corresponding to a shifted band of 14-3-3β as found by western blot. The bottom band most likely corresponds to the six 14-3-3 isoforms that have very similar sizes (α/β, ζ/δ, γ, σ, η and θ, all 245–248 AAs), while the top band probably corresponds to the ε isoform which is larger (255 AAs). The binding of the peptides to 14-3-3 was highly selective with virtually no other proteins significantly labeled in the lysate. Moreover, the bands for the methacrylate peptides intensify after a long incubation (22 h compared to 1 h) indicating their stability under these conditions.

We proceeded to test whether the peptides can detect 14-3-3 proteins in extracellular medium. We grew A549 cells in serum-free medium for either 24 h or 48 h, and then filtered the medium, concentrated it and exchanged the buffer. The peptides detected 14-3-3 in the medium with very high selectivity and sensitivity. In contrast to the results observed in lysates, 14-3-3σ was not detected by peptide **12** in the medium. This agrees with previous work that indicated that 14-3-3β is excreted in the medium while 14-3-3σ is not[86]. We also found that increasing concentration of the lysate during incubation does not affect the selectivity of the peptides (Supplementary Fig. 7B). Therefore, the methacrylate peptides **3** and **8** are powerful tools for the detection and quantitation of 14-3-3 isoforms in lysates and extracellular media.

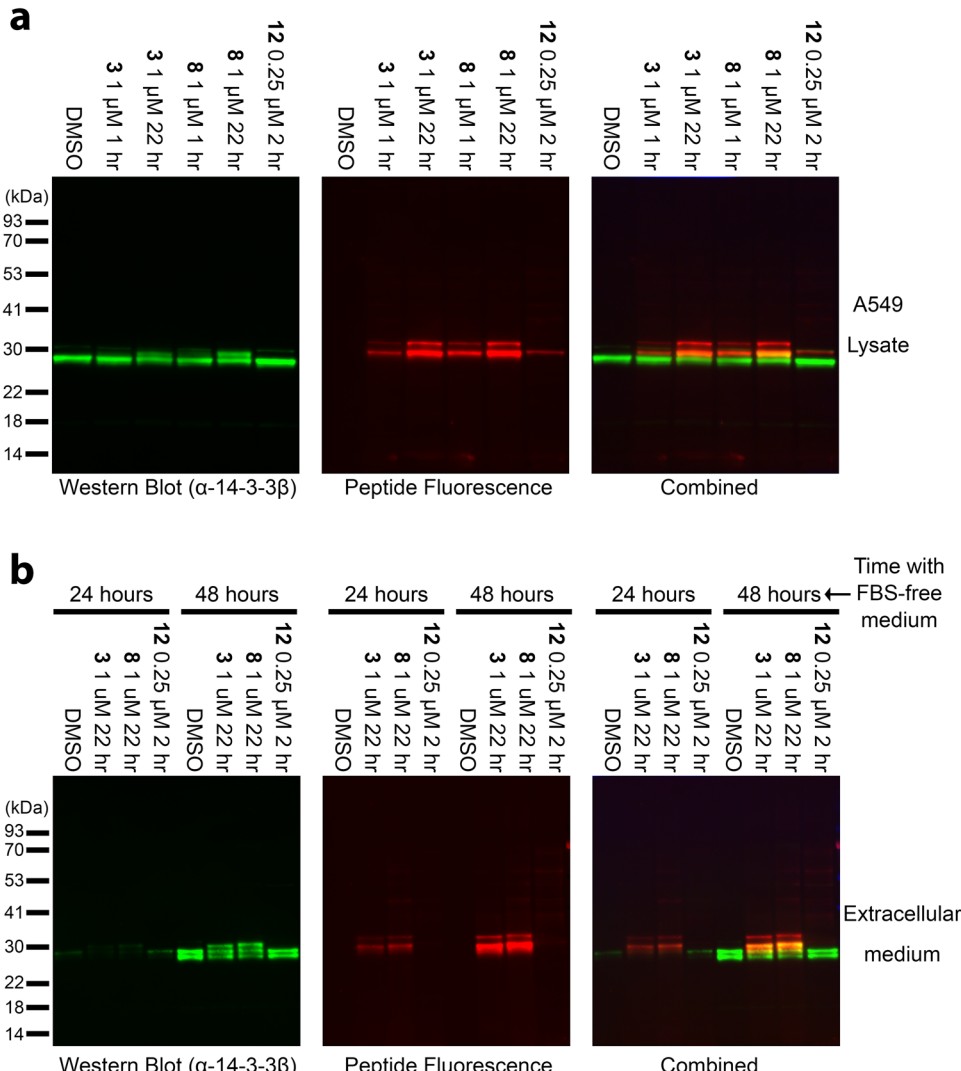

**Fig. 5 | Labeling of 14-3-3 proteins by BODIPY-modified peptides in A549 lysates and in medium.** A549 cells were grown for the indicated times in serum-free media. Lysates (**a**) or concentrated media (**b**) were then incubated for various times at room temperature with the peptides, followed by SDS-PAGE and western blot. Left Panel: Detection of 14-3-3β by western blot; Middle panel—peptide fluorescence; right panel—overlaid images. Experiments were repeated three times.

## Characterization of selectivity and off targets using chemical proteomics

To further characterize the selectivity and off targets of the methacrylate peptides, we synthesized biotinylated derivatives of peptides **3** and **8**, incubated A549 lysates with them, enriched the biotinylated proteins using streptavidin beads and used trypsin digestion followed by LC-MS/MS to characterize the bound proteins. All isoforms of 14-3-3 were bound efficiently and are the most prominent targets with few off-targets, confirming that peptides **3** and **8** were selective, pan-14-3-3 reactive probes (Fig. 6). Several off-targets were identified, many of which are NAD/NADP dependent enzymes such as aldo-ketoreductases, aldolases, and dehydrogenases. These contain a defined binding pocket for the phosphate containing cofactor with nearby lysine residues[87,88]. Enzymes with phosphate-containing substrates, including several glycolytic enzymes, were also prominent off-targets (Source data file). We speculate that the phosphorylated peptides may compete for these binding sites and form covalent adducts with these proteins. Nevertheless, the fluorescence imaging results indicate that 14-3-3 proteins are targeted very selectively, and that only a small fraction of the off-targets become modified due to lack of more specific sequence recognition.

## Development of electrophilic protein binders

Since this electrophile can be installed directly on non-modified peptides, we sought to modify a recombinant protein into a covalent binder using 2-(bromomethyl)acrylate. As a model system we selected the bacterial Colicin E9 toxin/anti-toxin system[79,80]. This system is composed of a highly toxic nuclease (E9) which is bound in the cell by an inhibitory partner termed the immunity protein (Im9). The complex can be excreted and internalized by target cells while displacing Im9, leading to E9-induced toxicity. The affinity of the Im9/E9 complex is very high and is well characterized structurally, which made it a promising starting point for the preparation of covalent binders.

We developed a computational pipeline that follows similar steps to those described in our peptide design protocol. However, instead of using CovPepDock to model the mutated complexes, we use the Rosetta Relax application[89,90], which performs all-atoms refinement using relatively small moves that sample the local conformational space, while applying covalent constraints between the methacrylate side-chain and the target lysine to enforce the covalent bond between them. Based on the non-covalent complex of colicin E9 and Im9 (PDB: 1EMV), we mutated 20 positions of Im9 that are within Cα-Cα distance <14Å from the target Lys97 to our methacrylate side-chain, and found five mutations that yielded sub-angstrom models (interface backbone

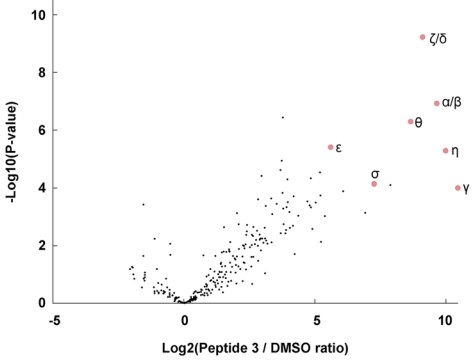
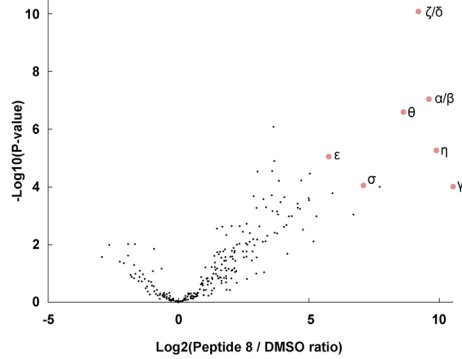

**Fig. 6 | Characterization of the selectivity of peptides 3 and 8 using pull-down proteomics.** A549 lysates were treated with biotinylated derivatives of peptides **3** or **8** (1 µM, 22 h 25 °C). The biotinylated proteins were enriched using streptavidin beads, digested with trypsin and analyzed using LC-MS/MS. P-values were calculated using a double-sided student's T-test.

RMSD < 1 Å) with particularly good interface and constraint scores (Supplementary Fig. 8). From these designs, we selected an Im9 mutant (C23A/E41C) onto which we would install a methacrylate 'warhead' to react with Lys97 in the E9 nuclease (Fig. 7a). We expressed and purified E9 and the Im9 mutants. Preparation of methacrylate-modified Im9 mutant under native buffer conditions was impractical as modification of the cysteine was slow and was competed by modification of other sites, as observed by the appearance of multiply-labeled species before full formation of the mono-labeled protein was observed, possibly indicating the cysteine was not fully exposed. Preparation under denaturing conditions (50% acetonitrile) was far more efficient, with rapid and selective modification in the time scale of minutes up to an hour, and combined with HPLC purification we obtained >95% single labeled protein (Supplementary Fig. 9).

To assess ligation of methacrylate-modified Im9 to Lys97 of E9, modified Im9 mutant and E9 were incubated and the cross-linked complex was monitored via intact protein LCMS. The results show about 50% conversion to the covalent complex within 5 h and near quantitative conversion within 16 h (Fig. 7b). To validate Lys97 ligation, a point mutation experiment was performed by individually mutating lysines 55, 81, 89, 97, and 125 in E9 to arginine. For the mutants K81R and K125R, bacterial growth was dramatically inhibited, possibly indicating reduction in the binding affinity leading to E9-mediated toxicity. We tested the binding K55R, K89R, and K97R to the methacrylate-modified Im9 mutant (Fig. 7c). While the K55R and K89R mutations had no effect on ligation efficiency, mutation of Lys97 abolished the formation of the covalent complex almost completely, indicating Lys97 as the target binding site, in agreement with the model.

Although the native binding affinity of these two proteins is very high, we wanted to test whether the mutation and covalent binding affects the structure and stability of the complex. To this end, we analyzed the pure Im proteins and the complexes with E9 using SEC-MALS (Supplementary Fig. 10). The estimated Mw of the proteins from the elution volume agree with the MALS measurements and indicated that Im proteins are monomeric, and that the C23A/E41C mutation in the Im protein makes the protein adopt a slightly more compact conformation, which may explain the difficulty of modifying the protein in native buffers. The methacrylate-modified mutant behaves more similarly to the WT, and the same trends are observed for the complexes with E9. These results indicate that the covalent complex adopts a similar structure to the native complex. To estimate the effect of the covalent binding on the complex stability, we used scanning differential scanning fluorimetry (DSF) to monitor the thermal stability of the complex (Supplementary Fig. 11). Im9 protein in its free and methacrylate-modified form exhibit unfolding around 50 °C, while the E9 protein does not show any discernible transition. While the non-covalent complex shows only minimal differences compared to free Im9, the covalent complex is considerably more stable, unfolding at 72 °C.

## Discussion

Covalent probes can react with target proteins with high potency and specificity and are becoming ever more useful as therapeutics and chemical biology tools[91]. However, several challenges remain for covalent agents, including for instance the fact that the majority of covalent probes target intracellular cysteine residues, and that small molecule covalent binders typically (although not exclusively) target traditional binding sites where a well-defined ligand binding pocket is present.

Peptides derivatized with covalent warheads can greatly expand the repertoire of addressable targets and are being increasingly explored[47]. We[55] and others[47] have shown that such electrophilic peptide binders could be designed to bind target proteins covalently and selectively and even repurposed to enable modifications such as fluorescent labeling, drug conjugation or conjugation to other proteins[92].

In this work, we developed an approach enabling the modification of native peptides or proteins with an amine/thiol-reactive electrophile. In the context of electrophilic peptides this approach is differentiated in two respects. First, is the facile synthesis that does not require non-canonical amino-acids and incorporates the electrophile in a single step on a native cysteine residue (Fig. 2), compared with somewhat more complicated syntheses and/or the use of expensive amino-acids reported for previous electrophilic peptides. An advantage of this property is that we envision such chemistry can be incorporated into covalent phage-display platforms[93] for efficient discovery of electrophilic peptides. Second, very few peptides were previously designed to irreversibly target lysine residues[47] and the few that did, utilized aryl sulfonyl fluorides[94–96]. To our knowledge, these are the first electrophilic peptides employing methacrylates to irreversibly label lysine target residues, as validated by crystallography (Fig. 4) and mass spectrometry (Supplementary Fig. 4).

We used our previously reported covalent peptide docking pipeline CovPepDock to design candidate peptides based on known binding partners of 14-3-3σ (Table 1). This yielded 3 irreversible binding peptides out of 11 candidates. The hit peptide **11** reacted with Cys38 rather than the predicted Lys122. The electrophile in **11** is indeed closer to Cys38 in the binding pocket compared to the lysine targeting **8** (Fig. 2a). The reaction of **11** with the cysteine appears to occur via two distinct mechanisms—addition and substitution, which were also observed when the peptide is incubated with cysteine. In addition, the cysteine adds via Michael addition to the methacrylate, while in substitution, the cysteine displaces the peptide, which is released as a free thiol, while the methacrylate remains on the protein. The

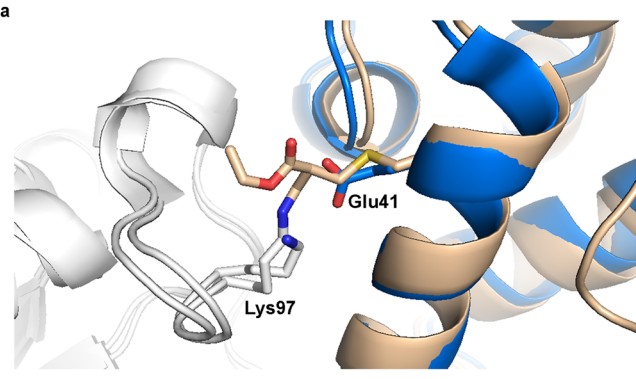

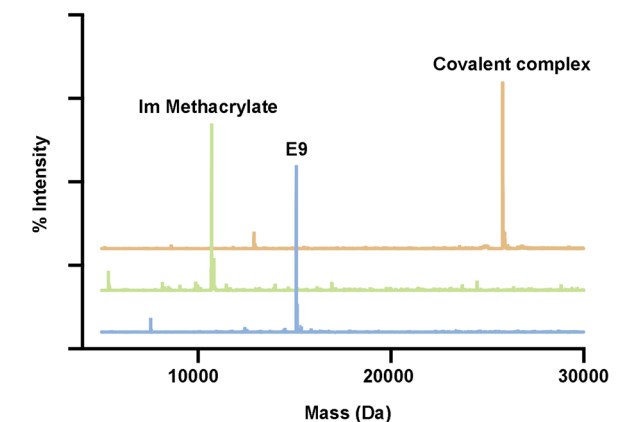

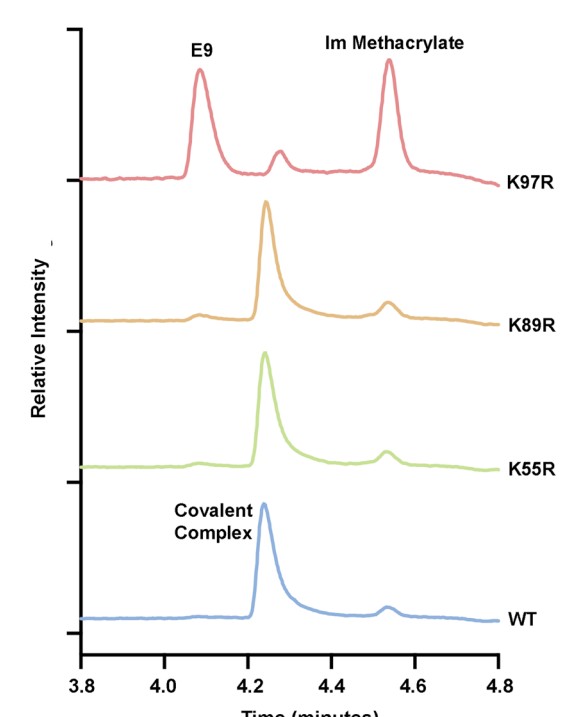

**Fig. 7 | Generation of an Im9 protein that can irreversibly bind E9. a** Model of the C23A/E41C mutant of Im9 (gold) covalently bound to Lys97 in E9 (white), compared to the wild type complex (with Im9 in blue). **b** Deconvoluted MS spectra of purified Im methacrylate, purified E9 and the covalent complex formed after their incubation. **c** Reverse phase HPLC chromatograms of samples of Im methacrylate incubated with wild type E9 and several E9 mutants. Incubation with the K97R mutant abolishes covalent bond formation.

methacrylate-labeled protein can later react via addition with free thiol peptide, eventually converting all the protein to an addition product, which is stable (Fig. 3b). Reaction of the methacrylate peptides with thiols such as cysteine and DTT also releases free peptide (Supplementary Figs. 3A, 4B). These results can be attributed to the higher nucleophilicity of the cysteine, which also reacts very rapidly with previously reported chloroacetamide-based peptides[55]. Nevertheless, the methacrylate-based peptides are not promiscuous and display high specificity towards the target protein (Figs. 5 and 6, Supplementary Fig. 7) and target residue as indicated by single labeling even with excess peptide (100-fold excess; Fig. 3a). None of the peptides reacted with Lysine 49, which may be attributed to the lower predicted nucleophilicity of this residue[83].

The potency and specificity of the peptides make them versatile chemical probes. While most known functions of 14-3-3 proteins are intracellular, they also serve some extracellular functions[86,97] and the presence of extracellular 14-3-3 proteins can serve as a biomarker for various diseases[97]. Therefore, the ability to bind and detect these proteins in extracellular media is of great interest. Using BDP-modified methacrylate peptides we managed to detect 14-3-3β in extracellular medium with sensitivity equivalent or higher than western blot (Fig. 5), illustrating the power of these probes. Furthermore, when targeting extracellular proteins, issues such as membrane permeability and proteolytic stability have less impact on the activity of the probe, opening the door for peptide and protein-based covalent probes.

Since the approach is applicable to peptides in their native form, it can be used to prepare electrophile-modified proteins directly from native, recombinant proteins. We demonstrated single labeling of Im9 with the methacrylate electrophile (Fig. 7, Supplementary Fig. 9). We should note that for the purpose of generating covalent protein reagents any number of cysteines can be mutated out, since the protein is generated by standard recombinant expression, supporting single installation of the electrophile. We then showed that the electrophilic Im9 could irreversibly bind E9 (Fig. 7b). The binding is abrogated, however, by mutation of the target lysine to an arginine (Fig. 7c), strongly supporting the designed binding mode. Such irreversible protein–protein binding can have significant effects, such as a very strong thermal stabilization as we show for the irreversible Im9/E9 complex (Supplementary Fig. 11) or improved in vivo efficacy as was reported for a covalent PD1/PDL1 complex[63]. By analogy to peptides, this approach would likely work even better for targeting cysteines on proteins. However, since the covalent protein reagents are not likely to be cell permeable, and most extracellular cysteines are oxidized[98] we speculate it would be found most useful for targeting lysines of extracellular targets.

Genetic code expansion has previously enabled the installation of fluorosulfates on proteins to target a histidine residue[63]. Here too, our approach offers a few advantages. First, genetic code expansion requires specialized bacterial expression systems and conditions that are not yet very widespread, somewhat limiting its broad applicability. Second, for future industrial applications such modified genetic systems might limit the scale of production, while canonical recombinant proteins with chemical modifications (such as Antibody-Drug-Conjugates[99]) were already proven applicable. Third, for genetic code expansion, extensive work should be undertaken to enable the installation of a new type of amino acid[100]. Thus, optimizing the features of the electrophile are limited. In our system, many electrophilic analogs of the bromo-methacrylate can be synthesized and conjugated to the same recombinant protein enabling much higher optimization throughput. Finally, the ethyl ester group can function not only as a point of diversification and screening for the development of better binders but also for various functionalization of the probes, for example via the attachment of E3 ligase binders for targeted degradation[101,102], for the attachment of fluorophores for imaging or

detection purposes[103], for targeting proteins to particular cell types or locations in the cell[104,105] and more.

Another aspect that contributes to the practicality of our approach is the computational modeling support. While manual inspection and selection of positions for introduction of the electrophile, may work well for a few peptides, automatic modeling and selection can cover larger number of possibilities (Supplementary Table 1). Moreover, as we expand the number of available electrophiles, with variable side-chains, modeling will be required to address the combinatorics (Supplementary Fig. 12). Finally, for *protein* covalent reagents (compared to peptides), manual inspection can be more challenging (Supplementary Fig. 13).

Our approach is not without its limitations. First, the modification of the peptide or protein into the methacrylate should be performed in the absence of reducing agents such as DTT or betamercaptoethanol, and preferably in the absence of high concentrations of amines such as tris buffer. Once the methacrylate peptide is synthesized, it is not prone to non-specific reaction with amines, but remains sensitive to thiols. While this could limit the use of these reagents in reducing environments such as cells, our results show the peptides retain activity in lysates even for prolonged incubations (Fig. 5). Furthermore, these reagents may be applied effectively in non-reducing environments such as extracellular media. A second issue is the possibility of an internal reaction between the introduced methacrylate group and a nucleophile such a lysine residue on the peptide or protein, which would inactivate it. However, the peptides did not react with high concentrations of lysine even after extended incubation, and the Im9 methacrylate remained fully capable of reacting with E9 despite the presence of surface lysines on the protein. Peptide **8** and **11** did react internally on the time scale of days, which is slow compared to the reaction with 14-3-3 proteins and did not interfere with labeling in lysates and in media (Fig. 5).

The most significant limitation is the relatively slow kinetics of covalent labeling in comparison to cysteine targeting probes. The lysine targeting peptides for instance reached 50% labeling within 2–3 h (Fig. 3b) compared to minutes for a cysteine targeting chloroacetamide peptide. Similar kinetics were observed with the E9/Im9 complex with 50% labeling at about 5 h. Such kinetics likely represent the low nucleophilicity of the target lysine, and are on a similar scale to the irreversible PD1/PDL1 complex (apparent second order rate ~890 $M^{-1}min^{-1}$)[63]. Similar issues with reaction rates arise with arylfluorosulfates[106–109], and the kinetics can be improved significantly by improving the positioning of the electrophile relative to the target residue[106]. Modulating the reactivity of the electrophile itself can also influence the reaction kinetics. We found that alteration of the ester substituent significantly affects the reactivity of the methacrylate electrophile. Preliminary results show that replacement of the ethyl ester with a phenyl ester dramatically increases the reactivity of the electrophile as well as the labeling rates of 14-3-3σ (Supplementary Fig. 12). Further studies of the stability and selectivity of these compounds, as well as testing of other ester substituents are currently ongoing.

In conclusion, we believe our approach offers a simple and versatile method for the preparation of a new class of covalent protein and peptide probes suitable to bind a large variety of biological targets, and as such would support new applications in chemical biology and covalent drug discovery.

## Methods

### Preparation of recombinant 14-3-3σ
A pPROEX HTb expression vector encoding the human 14-3-3σ with an N-terminal His$_6$-tag was transformed by heat shock into NiCo21 (DE3) competent cells. Single colonies were cultured in 50 mL LB medium (100 mg/ml ampicillin). After overnight incubation at 37 °C, cultures were transferred to 2 L TB media (100 mg/ml ampicillin, 1 mM MgCl$_2$)

and incubated at 37 °C until an OD600 nm of 0.8–1.2 was reached. Protein expression was then induced with 0.4 mM isopropyl-β-d-thiogalactoside (IPTG), and cultures were incubated overnight at 18 °C. Cells were harvested by centrifugation (17,000 × *g*, 20 min, 4 °C) and resuspended in lysis buffer (50 mM HEPES, pH 8.0, 300 mM NaCl, 12.5 mM imidazole, 5 mM MgCl$_2$, 2 mM βME) containing cOmplete™ EDTA-free Protease Inhibitor Cocktail tablets (1 tablet/100 mL lysate) and benzonase (1 μl/100 mL). After lysis using a C3 Emulsiflex-C3 homogenizer (Avestin), the cell lysate was cleared by centrifugation (60,000 × *g*, 30 min, 4 °C) and purified using Ni$^{+2}$-affinity chromatography (Ni-NTA superflow cartridges, Qiagen). Typically two 5 mL columns (flow 5 mL/min) were used for a 2 L culture in which the lysate was loaded on the column washed with 10 CV wash buffer (50 mM HEPES, pH 8.0, 300 mM NaCl, 25 mM imidazole, 2 mM βME) and eluted with several fractions (2–4 CV) of elution buffer (50 mM HEPES, pH 8.0, 300 mM NaCl, 250 mM imidazole, 2 mM βME). Fractions containing the 14-3-3σ protein were combined and dialyzed into 25 mM HEPES pH 8.0, 100 mM NaCl, 10 mM MgCl$_2$, 500 μM TCEP. Finally, the protein was concentrated to ~60 mg/ml, analyzed for purity by SDS-PAGE and Q-Tof LC/MS and aliquots flash-frozen for storage at −80 °C.

### Peptide synthesis
Reagents for peptide synthesis were purchased from Chem-Impex. Peptides were synthesized on Rink Amide resin using standard Fmoc chemistry on a 0.025 mmol scale. The resin was swelled for 30 min in dichloromethane (DCM), then washed with dimethylformamide (DMF). Fmoc deprotections were carried out using 20% piperidine in DMF (3 × 3 min), and couplings were performed as follows: 4 equivalents of amino acid were mixed with 4 equivalents of HATU/HOAT and 8 equivalents of DIPEA in DMF and added to the resin with mixing for 30 min. For phosphoserine and propargylglycine, 2 equivalents were used and reaction times were extended to 2 h. After the last Fmoc deprotection, the peptides were acetylated at the N terminus using acetic anhydride (10 equivalents) and DIPEA (20 equivalents) in DMF for 30 min. Finally, the resins were washed with DCM, dried in a desiccator, and cleaved using 94% TFA/1% DODT/2.5% TIPS/2.5% water for 2 h with tumbling. The cleaved peptides were precipitated in cold diethyl ether: hexane, washed once with ether, dried, dissolved in 50% acetonitrile and lyophilized.

The electrophile was introduced directly to the crude peptides as follows: Crude peptides were dissolved in 100 mM NaPi pH = 7.5 at a concentration of 25 mM. Ethyl 2-(bromomethyl)acrylate was dissolved in acetonitrile to 200 mM and 3 equivalents were added to the peptide solution. Reactions were monitored using LCMS and were typically complete within 1–2 h at room temperature. Reacted peptides were then purified using reverse phase HPLC.

To prepare fluorescently labeled peptides, a residue of propargylglycine was coupled to the peptide at the N-terminus prior to N-terminal acetylation, cleavage and reaction with Ethyl 2-(bromomethyl)acrylate. The pure peptide was then labeled as follows using copper-catalyzed azide–alkyne cycloaddition (CuAAC): 5 μl of 20 mM peptide was mixed with 15 μL of BDP-TMR azide (150 nmol). Water was added to 100 μL and about 50 μL tBuOH was added to dissolve the dye. At this point, CuSO$_4$:THPTA 100 mM (1 μL), and 200 mM sodium ascorbate (200 nmol, freshly dissolved) were added and the reaction continued for 1 h and the product was purified using HPLC.

The purity of all peptides was confirmed using LCMS.

### LC/MS instrumentation and runs
The LC/MS runs for 14-3-3σ were performed on a Waters ACQUITY UPLC class H instrument, in positive ion mode using electrospray ionization. UPLC separation used a C4-BEH column (300 Å, 1.7 μm, 21 mm × 100 mm). The column was held at 40 °C and the autosampler at 10 °C. Mobile phase A was 0.1% formic acid in water, and mobile phase B was 0.1% formic acid in acetonitrile. The run flow was

0.4 mL/min. The gradient used was 1% B for 2 min, increasing linearly to 80% B for 2.5 min, holding at 80% B for 0.5 min, changing to 20% B in 0.2 min, and holding at 1% for 0.8 min. The MS data were collected on a Waters SQD2 detector with an $m/z$ range of 2–3071.98 at a range of 900–1900 $m/z$. The desolvation temperature was 500 °C with a flow rate of 800 L/h. The voltages used were 1.00 kV for the capillary and 24 V for the cone. MassLynx version 4.2 was used to operate the LCMS and analyze the data. Raw data were processed using openLYNX and deconvoluted using MaxEnt with a range of 28,000: 34,000 Da and a resolution of 1 Da/channel.

The LS/MS runs for peptides were performed using the same instrument with a C18-CSH column (300 Å, 1.7 µm, 21 mm × 100 mm) using a gradient starting from 1% B for 1 min, rising to 95% B in 4.5 min, holding at 95% B for 0.75 min, then decreasing to 1% B in 0.75 min and holding at 1% B for 1 min. MS data were collected at a range of 80–2500 $m/z$, using identical conditions for ionization as with the protein.

## Binding experiments to 14-3-3σ
Peptide 100X stocks were prepared by dissolving in DMSO + 5 mM acetic acid and storing at −80 °C. Binding of peptides to 14-3-3σ was performed in 25 mM HEPES pH = 7.5, 100 mM NaCl, 10 mM MgCl₂. The protein was diluted to 2 µM in assay buffer, and the diluted protein was added to the peptide stock at 100:1 ratio and incubated in various conditions. For analysis, 24 µl sample was mixed with 6 µl of 2.4% formic acid in water and then 10 µl were injected to intact protein LCMS.

## Fluorescence polarization experiments
Fluorescence polarization experiments were performed using TECAN plate reader in dark 384-well plates in volumes of 50 µl in triplicates. The buffer was HEPES 25 mM pH = 7.5, 100 mM NaCl, 10 mM MgCl₂, 0.05% IGEPAL. For each sample, 0.5 µl of 100X stock of the competitor peptide (in DMSO + 5 mM acetic acid) was added, followed by 25 µl of 10 nM BDP-labeled non-covalent peptide probe[55]. Finally, 25 µl of 0.5 µM 14-3-3 σ was added to the plate and the plate was mixed. Polarization was measured at 27 °C.

## LC/MS/MS characterization of labeling sites of methacrylate peptides in 14-3-3σ
14-3-3σ was diluted to 2 µM in HEPES 25 mM pH 7.5, 100 mM NaCl, 10 mM MgCl₂, and incubated with 5 µM peptide in samples of 50 µl. The samples were incubated for 48 h at room temperature, resulting in ~75% labeling by peptide 3, 90% labeling by peptide 8 and 100% labeling by peptide 11. At this point 50 µl of 10% SDS in HEPES 25 mM pH = 7.5 was added and DTT was added to 5 mM, followed by incubation at 65 °C for 45 min. This was followed by addition of iodoacetamide to 10 mM and incubation of 40 min at room temperature in the dark. The samples were then processed using S-trap (Protify) according to the manufacturer's instructions, followed by desalting using Oasis plate (Waters).

Each sample was dissolved in 50 µl of 3% acetonitrile + 0.1% formic acid, and 0.5 µl was injected to the column. Samples were analyzed using EASY-nLC 1200 nano-flow UPLC system, using PepMap RSLC C18 column (2 µm particle size, 100 Å pore size, 75 µm diameter × 50 cm length), mounted using an EASY-Spray source onto an Exploris 240 mass spectrometer operated using Xcalibur version 4.4.16.14. uLC/MS-grade solvents were used for all chromatographic steps at 300 nL/min. The mobile phase was: (A) H2O + 0.1% formic acid and (B) 80% acetonitrile + 0.1% formic acid. Peptides were eluted from the column into the mass spectrometer using the following gradient: 1–40% B in 60 min, 40–100% B in 5 min, maintained at 100% for 20 min, 100 to 1% in 10 min, and finally 1% for 5 min. Ionization was achieved using a 1900 V spray voltage with an ion transfer tube temperature of 275 °C. Initially, data were acquired in data-dependent acquisition (DDA) mode.

MS1 resolution was set to 120,000 (at 200 $m/z$), a mass range of 375–1650 $m/z$, normalized AGC of 300%, and the maximum injection time was set to 20 ms. MS2 resolution was set to 15,000, quadrupole isolation 1.4 $m/z$, normalized AGC of 100%, and maximum injection time of 22 ms, and HCD collision energy at 30%. 3 injections of 0.5 µl were performed for each sample. The DDA data was analyzed using MaxQuant 1.6.3.4[110]. The database contained the sequence of the 14-3-3σ construct used in the study, and contaminants were included. Methionine oxidation and N terminal acetylation were variable modifications, and carbamidomethyl was a fixed modification in the analysis, with up to 4 modifications per peptide. Digestion was defined as trypsin/P with up to 2 missed cleavages. PSM FDR was defined as 1 and Protein FDR/Site Decoy fraction were defined as 0.01. Second Peptides were enabled and Match between runs was enabled with a Match time window of 0.7 min. The data was imported into skyline (version 22.2.0.351) and precursors from 9 peptides containing or following the residues Cys38, Lys49 and Lys122 were selected for parallel reaction monitoring (PRM). In every acquisition cycle, one full MS spectrum was taken at a range of 350–1000 Da, 300% AGC target, maximum injection time 20 ms at a resolution of 120,000. Data for each precursor was measured during a 4–5 min window around the retention time measured in the DDA run, with Q1 resolution of 2 Da, orbitrap resolution of 15,000, 300% AGC target and maximum injection time of 160 ms. The acquired data was then analyzed in skyline using a spectral library generated from the DDA runs. The 3 most intense product ions were used for quantitation relative to the DMSO control. data have been deposited to the ProteomeXchange Consortium via the PRIDE[111] partner repository with the dataset identifier PXD044257 and 10.6019/PXD044257.

## Crystallization of 14-3-3σ-peptide complexes
14-3-3s was C-terminally truncated (DC) after T231 to enhance crystallization. 14-3-3 and peptides 3/8 were dissolved in complexation buffer (20 mM HEPES pH 7.5, 100 mM NaCl, 10 mM MgCl₂) and mixed in a 1:2.5 or 1:5 molar stoichiometry (protein:peptide) at a final protein concentration of 10, 11, 12, and 12.5 mg/mL. The complex was set up for sitting-drop crystallization after overnight incubation at 4 °C, in a custom crystallization liquor (0.095 M HEPES (pH7.1, 7.3, 7.5, 7.7), 0.19 M CaCl2, 24–29% (v/v) PEG 400 and 5% (v/v) glycerol). Crystals grew within 5–10 days at 4 °C.

Crystals were fished and flash-cooled in liquid nitrogen. X-ray diffraction (XRD) data were collected at the Deutsches Elektronen-Synchrotron (DESY) PETRA III beamline P11, Hamburg, Germany.

Initial processing of datasets was done using CCP4i from the CCP4[112] suite. First, XIA2/DIALS[113] was run for data indexing and integration, and AIMLESS[114,115] for scaling. The structures were phased by molecular replacement, using protein data bank (PDB) entry 5N75 as a template, in MOLREP[116]. REFMAC5[117] was used for initial structure refinement. Correct peptide sequences were modeled in the electron density in Coot[117]. The presence of the covalent interaction of the peptides with Lysine 122 was verified by visual inspection of the Fo-Fc and 2Fo-Fc electron density maps in Coot and build in via AceDRG[118]. Finally, REFMAC5 and Coot were used in alternating cycles for model building and refinement. See Supplementary Table 2 for data collection and refinement statistics. The structures were submitted to the PDB with IDs: 8C2E and 8C2F. Validation reports are presented in the Source data file.

## Measurement of binding to 14-3-3 isoforms using LC-MS qTOF
The 14-3-3 isoforms were buffer-exchanged into complexation buffer (20 mM HEPES pH 7.5, 100 mM NaCl, 10 mM MgCl₂) and mixed with the peptides (3/8) in a 1:5 molar stoichiometry (protein:peptide) at a final concentration of 10 mg/mL. The complexes were buffer-exchanged into milliQ + 0.1% formic acid after overnight incubation at 4 °C.

UPLC-QToF-MS analysis was performed on a Waters (Milford, MA, USA) Acquity I-Class UPLC system coupled to a Waters Xevo G2 quadrupole time-of-flight (QToF) mass spectrometer. The devices were controlled by MassLynx Software (version 4.1, Waters, MA, USA). Full scan in positive electrospray ionization (ESI+) mode was used as MS acquisition mode with an acquisition range from 200–2000 $m/z$. A 3 μm, 150 × 2.0 mm Polaris 3 C8-A column (Agilent, Middelburg, the Netherlands was placed inside a column oven at 60 °C and used for chromatographic separation. Flow rate was set at 0.3 mL/min, and a gradient of water containing 0.1% (v/v) formic acid (A) and acetonitrile containing 0.1% (v/v) formic acid (B) was set as follows (all displayed as % v/v): 0.0–7.5 min (15% to 75% B), 7.5–8.0 min (75% B), 8.0–8.1 min (75% to 15% B), 8.1–10.0 min (15% B). Mass Spectrometry settings were set as follows: capillary voltage: 0.80 kV, cone voltage: 40 V, source offset: 80 V, source temperature: 100 °C, desolvation temperature: 400 °C, cone gas: 10 L/h desolvation gas: 800 L/h. The samples concentration were 0.01–0.1 mg/mL, and the injection volume was 1 μL. Deconvolution was performed by the MaxEnt1 option of the MassLynx software. Errors were calculated using the MaxEnt Errors option.

## Binding of 14-3-3 proteins to peptides in extracellular media and lysates

A549 cells were obtained from biological services of the Weizmann Institute. For experiments in lysates, A549 cells were grown in DMEM + FBS. The cells were washed with PBS, scraped from the plate and centrifuged 200 × $g$ for 5 min. The cells were lysed in HEPES 50 mM pH = 7.5, 150 mM NaCl, 1% IGEPAL with the addition of a protease inhibitor cocktail (Roche 11836170001) and phosphatase inhibitor cocktail (PhosStop from Roche, 1X). Cells were incubated in the lysis buffer and centrifuged at 21,000 × $g$ for 10 min at 4 °C. The protein concentration was estimated using BCA, and the lysate was diluted to 1.9 mg/ml in the lysis buffer. For incubation with the peptides, 38 μl of lysate was mixed with 2 μl of 20 X stock of the peptide (for peptides **8** and **3**: 20 μM in 20% DMSO/lysis buffer; for peptide **12**: 5 μM in 20% DMSO/lysis buffer; for no peptide: 20% DMSO/lysis buffer) and incubated at 25 °C in the dark. Then, 13.3 μl of 4X LDS samples buffer with 20 mM DTT was added and the samples were heated for 10 min at 70 °C. The samples were loaded on Bis-Tris gradient gels (4–20%, Genscript) and run using Tris-MOPS buffer at 60 mA/200 V. The gels were transferred to nitrocellulose membrane and the membrane was blocked with 5% BSA/TBST for 1 h RT. The membrane was incubated overnight at 4 °C with 1:500 diluted anti-14-3-3β antibody (abcam ab15260) in 5% BSA/TBST. The membrane was washed thrice with TBST and incubated with 1:2000 diluted anti-rabbit HRP antibody (CST 7074S) for 1 h RT in 5% BSA/TBST. The membrane was washed thrice with TBST and imaged as followed: fluorescence using 546 nm excitation was measured using ChemiDoc (BioRad) using 9 s exposure, and chemiluminescence was measured using 20 s exposure. Images were processed and generated using ImageLab.

For experiments in medium, after growing the cells they were transferred to DMEM without FBS, followed by incubation for either 24 h or 48 h. After the incubation, 8.5 ml of the medium was filtered through a 0.2 μm filter, concentrated using a centrifugal concentrator (vivaspin, cutoff 8000–10,000 Da) to ~200 μl, and diluted to 8 ml using HEPES 50 mM pH = 7.5, 150 mM NaCl. This was followed by 2 additional rounds of dilution and concentration to ~200 μl. The samples were then diluted to 300 μl with IGEPAL added to 1% as well as protease inhibitors and PhosStop. 38 μl samples from the medium were then incubated with the peptides and analyzed as performed for the lysate, with 15 s exposure for fluorescence and 50 second exposure for chemiluminescence.

For experiments in which the gel was directly imaged, after the run the gel was immersed in fixation solution (45% methanol, 45% water and 10% acetic acid) for 10 min, followed by 2 washes with Tris 100 mM pH = 8 in water. Afterwards the gel was directly imaged in ChemiDoc.

## Pull down proteomics experiment

We synthesized and purified N terminally biotinylated derivatives of peptides **3** and **8**. A549 cells were harvested and lysed as described before. The lysates were diluted to 1.9 mg/ml in lysis buffer, and the peptides were diluted to 20 μM in 20% DMSO/lysis buffer. 142.5 μl of lysate were mixed with 7.5 μl of 20 μM peptide stock and the samples were incubated at 25 °C for 22 h. The proteins were precipitated by addition of 450 μl water, 600 μl HPLC grade methanol and 150 μl HPLC grade chloroform, followed by vortexing and centrifugation for 10 min at 21,000 × $g$ at 4 °C. The top layer was aspirated, and 600 μl methanol was added and the sample was vortexed and centrifuged again, followed by aspiration of the supernatant. The pellet was air-dried and kept at −80 °C. The pellet was dissolved in 200 μl of 2.5% SDS in PBS with heating to 60 °C and shaking at 1150 rpm for 30 min. After dissolution, the sample was diluted 20-fold in PBS and incubated with 10 μl of streptavidin agarose beads (Thermo) with tumbling for 3 h at room temperature.

The beads were then filtered through spin columns in a vacuum manifold. The beads were washed twice with 1% SDS/PBS (300 μl), dispersed in 300 μl 1% SDS/PBS and 3 μl of 1 M DTT were added with 30 min incubation at room temperature. Then, 15 μl of freshly dissolved 0.8 M iodoacetamide were added followed by 30 min incubation at room temperature in the dark. The solution was then removed and the beads were washed three times with 350 μl of freshly dissolved 6 M urea in PBS, three times with 400 μl of 20% methanol in PBS, once with PBS and twice with water. The beads were then transferred to tubes using 100 μl of 50 mM triethylammonium bicarbonate, and the bound proteins were digested with 0.5 μg of trypsin (promega) at 37 °C with shaking at 1200 rpm for 6 h.

The beads were centrifuged and the supernatant was mixed 1:1 with 0.2% TFA in water. The peptides were desalted using Oasis desalting columns (Waters) and dried under vacuum. The dry peptides were dissolved in 3% acetonitrile + 0.1% formic acid (25 μl) and 2 μl were injected. Samples were analyzed using EASY-nLC 1200 nano-flow UPLC system, using PepMap RSLC C18 column (2 μm particle size, 100 Å pore size, 75 μm diameter × 50 cm length), mounted using an EASY-Spray source onto an Exploris 240 mass spectrometer. uLC/MS-grade solvents were used for all chromatographic steps at 300 nL/min. The mobile phase was: (A) $H_2O$ + 0.1% formic acid and (B) 80% acetonitrile + 0.1% formic acid. Peptides were eluted from the column into the mass spectrometer using the following gradient: 1–40% B in 160 min, 40–100% B in 5 min, maintained at 100% for 20 min, 100 to 1% in 10 min, and finally 1% for 5 min. Ionization was achieved using a 2100 V spray voltage with an ion transfer tube temperature of 275 °C. Initially, data were acquired in data-dependent acquisition (DDA) mode. MS1 resolution was set to 120,000 (at 200 $m/z$), a mass range of 375–1650 $m/z$, normalized AGC of 300%, and the maximum injection time was set to 20 ms. MS2 resolution was set to 15,000, quadrupole isolation 1.4 $m/z$, normalized AGC of 50%, automatic maximum injection time, and HCD collision energy at 30%. Four samples were analyzed per condition.

Data analysis was performed using Fragpipe (version 19.1) using Msfragger search engine (version 3.8)[119,120], IonQuant 1.8.10[121], and Philosopher 4.8.1[122]. Analysis was performed using a human proteome database from December 2022 (Uniprot) with contaminants added and with Streptavidin added manually as a contaminant. Msfragger analysis was performed using Trypsin as the enzyme that cuts after Arg and Lys, with up to 2 missed cleavages, peptide length 7-50 and the N terminal methionine removed. N terminal acetylation and methionine oxidation were defined as variable modifications and carbamidomethyl was defined as a fixed modification. False discovery rate of 0.01 was used both at the peptide and the protein level. Label-Free Quantification was performed using IonQuant, with Match Between Runs enabled with a tolerance of 1 min. After analysis, the combined protein file was analyzed using Perseus[123]. Intensities were converted to Log2

values, the quadruplicates of each type were grouped, and all proteins for which there were at least 3 valid values in one of the groups were kept in the analysis. Missing values were replaced by imputation from a normal distribution (downshift 1.8, width 0.3), and differences and *P*-values were calculated using double-sided student's t-test. The data have been deposited to the ProteomeXchange Consortium via the PRIDE partner[111] repository with the dataset identifier PXD044294.

### Cloning of Im9 mutant and E9 mutants

pET21d plasmids encoding for either E9 + wild type Im9 or for Im9 alone were a gift from the lab of Sarel Fleishman in the Weizmann Institute. For mutation of E41 into cysteine, we performed PCR using the plasmids as a template using the following primers:

    ImFor: GAAATGACTGAGCACCCTAGT
    ImRev: ACAAAAGTGTGTAACCAATTTAACCAGTTC

The PCR product was purified and 1 μg was phosphorylated using 10 units T4 PNK (NEB) in 20 μl of T4 ligase buffer (NEB) for 1 h at room temperature. This was followed by addition of 400 units of T4 ligase (NEB) for 2 h at room temperature. The product was transformed to DH5α and plated on ampicillin plates. After pick of colonies and identification of correct sequences, this step was repeated with the following primers to introduce the second mutation (C23A):

    Im2For: GCTAATGCGGACACTTCCAGTG
    Im2Rev: AATTGTTGTTACAAGCTGTAAAAATTCAG

For mutations of E9 we used the same procedure with the following sets of primers:

    Lys55for: CGGGCTGTATGGGAAGAGGTGTC
    Lys55rev: CCGAAAATCGTCGAAGCTTTTAAATTC
    Lys81for: CGAGGTTATTCTCCGTTTACTCCAAAG
    Lys81rev: TGAAACACTAGACTTATTGCTTGGG
    Lys89for: CGGAATCAACAGGTCGGAGGG
    Lys89rev: TGGAGTAAACGGAGAATAACCTTTTG
    Lys97for: CGAGTCTATGAACTTCATCATGACAAG
    Lys97rev: TCTCCCTCCGACCTGTTG
    Lys125for: CGGCGACATATCGATATTCACCG
    Lys125rev: AGGTGTAGTCACTCGGATATTATC

### Expression and purification of E9 and Im9 mutants

The plasmids were transformed into BL21(DE3) bacteria. The bacteria were grown in 2YT + NPS + 1 mM MgSO₄ at 37 °C to OD = 0.6, cooled rapidly on ice to 16 °C, and induced using 1 mM IPTG for 16 h.

For purification of Im9, the cells were dispersed in 30 ml of lysis buffer (Tris 25 mM pH = 7.5, 50 mM NaCl, 10 mM imidazole) + protease inhibitors, and sonicated (55%, 1 min, 5 s pulses). After this, MgCl₂ was added to 1 mM and 5 μl of benzonase nuclease (Fischer) were added. The lysates were spun (60,000 × *g* for 20 min), and the lysates were filtered 0.45 μm. Then, each lysate was loaded on Ni-NTA column (5 ml) preequilibrated with lysis buffer, and the column was washed with 4 CV of lysis buffer. Im9 was eluted with Tris 25 mM pH = 7.5, 50 mM NaCl, 500 mM imidazole, dialyzed extensively against NaPi 20 mM pH = 7.2, 100 mM NaCl (3 times), filtered 0.2 μm and flash frozen in −80 °C.

Purification of E9 was performed using an identical procedure, except that elution was performed using 6 M GuHCl. Some precipitation was observed during dialysis.

### Methacrylate labeling of Im9 mutant

Labeling was performed in protein storage buffer (NaPi 20 mM pH = 7.2, 100 mM NaCl). Im9(C23A/E41C), at a concentration of 1.88 mM, was diluted two-fold in acetonitrile, leading to some precipitation. At this point 1.1 equivalents of ethyl-(3-bromomethacrylate) (dissolved beforehand in acetonitrile) were added. After 1 h at room temperature, 70% labeling was observed, and another 0.7 equivalents were added. After an hour the sample was diluted in 0.1% TFA in water, filtered, and purified using HPLC.

### Reaction between Im9 methacrylate and E9

The purified proteins were diluted to 20 μM in NaPi 20 mM pH = 7.2, 50 mM NaCl. Then the Im methacrylate solution was mixed in a 1:1 ratio with the E9 solution, giving 10 μM complex. The reactions were incubated at room temperature for 4.5 h, and then stopped by diluting the complex 5-fold in 0.1% TFA/water, followed by LC-MS analysis.

### Sec-MALS characterization of Im-E9 complexes

Samples containing 200 μM isolated Im constructs or Im-E9 complexes were prepared in NaPi 25 mM pH = 7.2, 100 mM NaCl. A miniDAWN TREOS multi-angle light scattering detector, with three angles (43.6°, 90°, and 136.4°) detectors and a 658.9 nm laser beam, (Wyatt Technology, Santa Barbara, CA) with a Wyatt QELS dynamic light scattering module for determination of hydrodynamic radius and an Optilab T-rEX refractometer (Wyatt Technology) were used in-line with size exclusion chromatography analytical Superdex 75 Increase 10/300 GL column (Cytiva). 420–770 μg of each sample were injected to the column in 150–200 μl. Experiments were performed using an AKTA Pure system with a UV-900 detector (Cytiva), at flow rate of 0.8 mL/min and with PBS pH = 7.4 as the running buffer. All experiments were performed at room temperature (25 °C). Data collection and SEC-MALS analysis were performed with ASTRA 6.1 software (Wyatt Technology). The refractive index of the solvent was defined as 1.331 and the viscosity was defined as 0.8945 cP (common parameters for PBS buffer at 658.9 nm). dn/dc (refractive index increment) value for all samples was defined as 0.185 mL/g (a standard value for proteins).

### Differential scanning fluorimetry for Im-E9 complexes

50 μl samples of E9:Im complexes in a concentration of 50 μM were prepared and incubated overnight at room temperature in NaPi 20 mM pH = 7.2, 50 mM NaCl. SYPRO Orange (X5000 stock) was diluted 200-fold in buffer, and from this stock 13 μl were added to each sample, diluting the protein to 40 μM. Each sample were split into 3 technical replicates and heated in a thermal cycler over 1.5 h to 95 °C while measuring the fluorescence.

### Introducing new residues to Rosetta

Our methacrylate side-chain was introduced to Rosetta using the protocol described in Renfrew et al.[124]. As the reaction between the methacrylate warhead and the lysine amine forms two different stereoisomers, we implemented both of them as different residues. We used the GaussView interface to draw each stereoisomer, and then used the Gaussian software to optimize the structures, with the following options: HF/6-31G(d) scf = tight test. Each optimized structure was converted to a mol file using OpenBabel toolbox (http://openbabel.org), and then to a Rosetta residue 'params file' using the molfile_to_params_polymer.py script provided in Rosetta. To allow the residue to form a covalent bond to another residue, we added a CONNECT record to each stereoisomer params file, specifying which atom participates in the inter-residue covalent bond, as described in Drew et al.[53] for oligooxopiperazines. We also added a virtual atom to each params file, and defined its internal coordinates according to the optimal position of the lysine NZ atom as predicted by the Gaussian optimization. These virtual atoms are used during the modeling process to favor the correct covalent bond geometry. Rotamer libraries were generated using the Rosetta MakeRotLib application.

A suitable covalently-linked variant of lysine was implemented through the residue patch system, to utilize the existing definitions and rotamer libraries that have been optimized for use in Rosetta[125]. We modeled the reacted lysine as described above, and created a patch file that deletes the 3HZ atom of lysine, and adds a CONNECT record and a virtual atom with internal coordinates that match the Gaussian optimized structure. We also added a PROTON_CHI record to allow sampling of the new rotamers around the bond CE-NZ bond.

## Design of 14-3-3σ peptide binders

We used PDB ID: 3MHR as a template structure to design Lys49- and Lys122-binding peptides for 14-3-3σ. We used Rosetta fixed backbone design application (fixbb) to mutate each lysine to our covalently-linked variant, and the relevant peptide positions (Cα-Cα distance to the target lysine <14 Å) to each of our methacrylate side-chain stereoisomers; these include positions 126–131 for Lys49 and positions 126-133 for Lys 122. We then applied CovPepDock to generate 200 models of each of these mutated complexes (100 for each stereoisomer). To favor the formation of the covalent bond in its correct geometry, we applied AtomPair constraints between each of the covalent bond atoms and its virtual placeholder in the partnering residue, as described in our previous work[55]. We used the HARMONIC score function, centered at 0 and with a standard deviation of 0.3. We manually inspected the 10 top-interface-scoring models of each complex, focusing on near-native models with constraint score <2, and selected 4 high-ranking peptides.

For our second set of peptides, we searched the PDB for X-ray crystal structures of 14-3-3σ in complex with a 3–15 amino acids long peptide. We then filtered the results for structures where the peptide binds near Lys122 (Cα-Cα distance <14 Å) but not near Cys38 (Cα-Cα distance >12 Å). This yielded the PDB IDs 3IQU, 3P1N, 4IEA, 4QLI, and 7NWF. Similarly, we designed Lys122-binding peptides for 14-3-3σ based on each of these structures, by mutating positions 257–260 of 3IQU, 372–374 of 3P1N, 620–625 of 4IEA, 175–180 of 4QLI and 592–595 of 7NFW. The native Cys180 of the 4QLI peptide was mutated to serine, to avoid the possible cyclization or side-reactions which may occur due to the addition of the second cysteine onto which we would install the methacrylate warhead.

## Design of Colicin E9 Protein Binders

We used PDB ID: 1EMV as a template structure. Similar to our peptide design protocol, we used Rosetta fixed backbone design application (fixbb) to mutate Lys97 of colicin E9 to our covalently-linked variant, and to mutate positions 30–41 and 48–55 of Im9 to each stereoisomer of our methacrylate side-chain. We then used the RosettaScripts interface and the FastRelax mover to generate 200 models of each complex (100 for each stereoisomer), while applying similar constraints to these described in the peptide design method section. To select a construct for synthesis and testing, we manually inspected the 10 top-interface-scoring models of each mutated complex, focusing on near-native models with constraint score <2.

## Reporting summary

Further information on research design is available in the Nature Portfolio Reporting Summary linked to this article.

## Data availability

Crystal structures have been deposited to the protein data bank (PDB) using IDs 8C2E and 8C2F. Proteomics data was deposited to the ProteomeXchange Consortium via the PRIDE partner repository with the dataset identifiers PXD044257 and PXD044294. The rest of the data presented in this manuscript is given in the Source data file. Source data are provided with this paper.

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

## Acknowledgements

N.L. would like to acknowledge funding from the Israel Science Foundation (grant no. 2462/19), The Minerva Foundation, The Israel Cancer Research Fund, and the Moross Integrated Cancer Center. N.L. is also supported by the Honey and Dr. Barry Sherman Lab, Dr. Barry Sherman Institute for Medicinal Chemistry, Nelson P. Sirotsky, the Goldhirsh-Yellin Foundation and Celia Zwillenberg-Fridman. M.O. is supported by the NWO OCENW.KLEIN.300 10028177 project. The plasmids for expression and purification of E9 DNAse and immunity protein were a gift from the laboratory of Sarel Fleishman in the Weizmann Institute. We acknowledge DESY (Hamburg, Germany), a member of the Helmholtz Association HGF, for the provision of experimental facilities. Parts of this research were carried out at PETRA III, and we would like to thank Guillaume Pompidor for assistance in using beam P11. Beamtime was allocated for proposal I-20211300 EC.

## Author contributions

N.L. conceived the research; R.G. synthesized the peptides, expressed the Im9 and E9 proteins, performed binding and proteomics experiments, analyzed the data, and wrote the manuscript; B.T. designed the covalent docking pipeline, and performed the modeling and design of peptide sequences; R.N.R. participated in the design of the chemical approach; M.V.D.O. performed the crystallography, structure determination, and binding experiment to 14-3-3 isoforms; H.A. performed SEC-MALS measurements; P.J.O., C.O. and N. L. participated in writing and revising the manuscript.

## Competing interests

R.G., R.N.R., B.T. and N.L. are inventors on a provisional patent application describing this technology. All other authors declare no competing interests.
