## [Peer Review File · Nature Communications]

A simple method for developing lysine targeted covalent protein reagentsREVIEWER COMMENTS

Reviewer #1 (Remarks to the Author):

This is a novel and interesting idea to develop covalent probes to target simultaneously Cys or Lys residues on protein surfaces. The approach is synthetically simple, and the results have been reported with a good level of detail. A few questions remain, as the authors noted, relative to the reactivity of the agents towards Lys but also Cys, with kinetics that is apparently very slow. One question to be addressed is the aqueous and plasma stability of the agents (the experiments with the lysates would suggest some stability but it would be nice to have those quantifications given that incubation times are very long: overnight or many hours). It also may be also useful to compare the proposed methacrylates to fluorosulfates (the authors cite correctly sulfonyl fluorides which are more reactive – but fluorosulfates are very stable and less reactive). Aryl-fluoro-sulfates suffer from the same slow kinetics but react more readily when properly juxtaposed to a binding site targeted Lys (see for example Baggio et al. *J. Med. Chem.* 2019, 62, 20, 9188–9200). Hence, the main limitation being the relatively slow kinetics of covalent labeling in comparison to cysteine or Lys targeting probes may also be due to the chosen pairs, and this could be discussed in the manuscript. There is a need for novel synthetically accessible electrophiles to expand the druggable space available to covalent ligands, hence I feel this current manuscript supports new applications for the design of covalent chemical probes and pharmacological tools.

Reviewer #2 (Remarks to the Author):

The manuscript by Gabizon et al describes the installation of a methacrylate warhead onto peptides and proteins to create covalent binding ligands for target proteins. This paper, following their 2021 *JACS* paper (doi.org/10.1021/jacs.0c10644) on methacrylamide-based warheads and their *ChemSci* paper on covalent peptide docking (CovPepDock, DOI: [10.1039/d1sc02322e](https://doi.org/10.1039/d1sc02322e)), seeks to develop methodologies to incorporate methacrylate into known peptide and miniprotein ligands to bind the corresponding target proteins. Specifically, the paper reports an alpha-bromomethyl derivative of acrylate, which couples with a cysteine thiol engineered into ligand peptides/proteins to achieve warhead incorporation. The exact peptide sequence and position of warhead position were determined with the assistance of computational modeling via CovPepDock. Two successful example of covalent peptide/protein ligands are described. This contribution addresses an important topic, that is the rational development of peptide/protein-based covalent inhibitors. As the CovPepDock methodology and the use of methacrylamide to target cysteines have both been reported, the novelty of this paper lies in the use of methacrylate to target lysines, for which solid evidence is provided with a crystal structure. The paper would be of interest to the community of covalent inhibitor development. However, the following two issues should be addressed before publication:

1. Target specificity should be better characterized. The peptide fluorescence gel (Supplementary Figure 3) shows additional bands, what are those proteins targeted by the covalent peptide? The effect of lysate concentration should be examined? It seems that only one concentration was assessed and the exact concentration was not specified. The target specificity of the peptide covalent inhibitors should be examined at high lysate concentrations to mimic intracellular environments.
2. The advantage of CovPepDock is not obvious. The authors identified residues for mutation/warhead incorporation based on the distance to the targeted residue. That can be with just PyMol examining the peptide-protein complex structure. More details should be given on CovPepDock results, how many peptide variants were examined in total computationally? What are the non-obvious details revealed through this docking program that eyeballing the complex structure would not give?

Reviewer #3 (Remarks to the Author):

The work done by Gabizon et al. presents a simple bioorthogonal strategy to proximity ligate peptides or even full-length proteins to their interaction partners with the help of ethyl 2-(bromomethyl)acrylate, a cheap reagent available commercially. Overall, I found their work interesting scientifically and their simple approach attractive over other methods that are out of reach for most research teams. I highly recommend the work for publication after minor modifications.

Questions related to text:

- 1, It may be a very naive question from a chemistry point of view, but can the ethyl 2-(bromomethyl)acrylate react with Lys residues of a peptide, or it will always preferably react with Cys? Will it react with a Cys-free peptide that contains a single Lys? How much weaker is the reactivity of the acrylate group to Cys? Is it possible that a side reaction could occur where the bromomethyl remains intact and the peptide is labeled with acrylate? I think for non-chemists, these discussions could come useful. The authors propose a versatile approach that is not limited to use by chemists, but the same community mostly will be unaware of the practical considerations of the approach.
- 2, Although the authors avoided using primary amine-containing buffers, such as the most commonly used TRIS, but they do not mention it in their work. I think it could be useful to mention if this reaction could work, or not in TRIS buffer. Also, the authors use bME during their purification steps. Is residual bME contamination a concern?
- 3, What ensures that the warhead will not intramolecularly? Only one of the tested peptides contained an intramolecular Lys residue (peptide 7) and this peptide did not show a reaction with 14-3-3 sigma. What is the author's opinion point regarding this issue? How likely that the warhead could staple the peptide intramolecularly turning a good potential binder inactive?

What happens if the peptide has a free N-terminus?

4, Naively looking, peptides with modification at the P+3 position should target this Lys, but none of them reacted with 14-3-3s significantly. Lys49 is involved in phosphate binding. Is it realistic to expect a reaction to this site?

5, The authors show that the reaction forms relatively slowly, but I have found some discrepancies in their data. In their first screen with their peptide panel, they performed the experiment with 100-fold molar excess overnight (approximately 14-16h) resulting 30-60% reaction. Then, they perform an experiment with 2.5-fold molar excess for 5h reaching >80% reaction. By decreasing the concentration of the peptide, I would expect a slower reaction. I think the key behind the observed reaction acceleration is that this reaction is very temperature sensitive and while the first reaction was performed in cold, the second was performed at room temperature. If this is true, how fast could be this reaction at more physiological temperatures (37 degrees)? Given the differences between the reaction rates at different temperatures, the authors should specify in the main text the conditions at each reaction and not only in the method section.

6, For such a slow reaction, it could be possible to measure the dissociation constants of the peptide inhibitor, as well as the reaction rate constant. I think it could be interesting to discuss how labeling the peptide could change its biophysical properties and for example, decipher why only 3-4 peptides out of 11 showed a reaction with 14-3-3 sigma. Is it possible that the other peptides only bind with very weak affinities after labeling?

7, The authors mention that the adduct is unstable in the presence of reducing agents. This is concerning because most proteins require reducing agents in vitro and because the cellular environment is reducing. What is the expected half-life of this bond, can it be suitable as is in live cell assays?

8, The R_{free} of the structure with peptide 8 seems to be high compared to the resolution, as well as the gap between R_{work} and R_{free} is high. I would be curious if the authors could comment on what limited their refinement in this instance.

9, Are there any additional Lys or Cys residues that have excess electron density in the structures?

10, It is a great idea to create 14-3-3 inhibitors that target the whole family instead of a particular isoform. I am curious why the authors choose sigma as their primary target, which appears to be the most divergent out of the family that also appears to bind with markedly weaker affinity to all targets. Would it be possible that the other peptides would react with higher efficiency with other isoforms, such as gamma, or eta?

11, I tend to agree with the authors that the BODIPY-modified peptides preferably target 14-3-3s in cell extracts (or in the medium). However, I have some problems with the way how the

supporting data is presented. First, specificity cannot be decided based on the cropped images shown in the main figure, because a highly specific off-target can run at a different MW. It would be better to show the entire lanes. Second, the markers need to be labeled better. Either units or complete annotations are missing. In my opinion, multiple bands are labeled with all peptides based on the fluorescence data that shows that neither of these peptides are fully specific. I do not think this is a problem, a full range of other molecules are expected to interact with such molecules such as phosphatases, kinases, or others, but the authors should be more straight with their interpretation.

12, Since the 14-3-3 beta antibody stains multiple bands, I wonder if we can consider this as a specific staining. Given the close homology between the 14-3-3 isoforms, I would not be surprised if we would see a pan 14-3-3 pool in the blot. Also, how many times the experiment was repeated?

13, The fact that denaturing of Im9 was essential for labeling is very unfortunate if the authors would like to demonstrate that the labeling approach is compatible with native conditions. Based on the structures, E41 appears to be solvent exposed. Is it possible that the protein forms some intrinsic oligomers that hide this residue? If yes, is it possible that the labeling prevents the formation of the same oligomeric state? I would be curious to see SEC data on these proteins, or CD spectra before and after labeling that could show that neither the oligomeric state Im9, nor the overall conformation of the protein is affected by the chemical labeling.

14, Regarding the thermal stability experiments, I do not find the results particularly interesting from the method point of view and the results are not very surprising. Complexes tend to be more stable than their parts and a covalent complex is expected to be more stable than a transient one. In my opinion, instead of the stability assay, it would have been more interesting to show that the overall conformation of the transient and covalent complex is practically the same with CD spectroscopy or SAXS, or other assays. I would propose moving Figure 7 to the supplement.

15, The authors should check subscripts and superscripts in their chemical formulas. (E.g. Ni²⁺, MgCl₂, CaCl₂...)

Questions related to figures:

Figure 2D - I get the idea that it is a schematic figure representing the major steps, but I find the lack of sequence disturbing. Is there only a single Cys in the peptide? Are there Lys residues? If you would like to show real data, please include the sequence, or only show schematic data.

Figure 3 - I find it difficult to interpret the intermediate complex of peptide 11. Was free ethyl 2-(bromomethyl)acrylate still present at this stage of the experiment? Was not the peptide HPLC purified after labeling? Or the linkage between the warhead and the peptide is fragile and can be broken resulting in a free acrylate compound that could reversibly react with the protein?

Figure 3 - Peptide 1 shows two bands that are not commented on in the text. Are these meaningful products, too?

Figure 4 - What is "electron density"? Is that a final 2FoFc map at a certain sigma level? Or an omit map? The authors need to be more precise on this point. In case this is a final map, the authors should show a bias-free omit map (e.g. simulated annealing omit map) in the supplement.

Figure 4 - Why Lys49 is shown in the figure?

Figure 6C- I get that the authors show a chromatogram, but what sort of chromatogram? Gel filtration, reverse phase HPLC, or something else?

SF2C - What are 3, 8, 12? Peptide IDs, molecular weights, time, or temperature? Clarify labels better in general, deciphering figures should be less guesswork.

Response to reviewers

A simple method for developing lysine targeted covalent protein reagents

We would like to thank the reviewers for their thorough evaluation of our manuscript. We have extensively revised the manuscript based on the reviewers' comments and performed several additional experiments to address their concerns. We believe the manuscript was greatly improved based on their feedback. Please find below a point-by-point answer to the reviewers' comments. Our answers are in red for clarity.

Reviewer 1:

This is a novel and interesting idea to develop covalent probes to target simultaneously Cys or Lys residues on protein surfaces. The approach is synthetically simple, and the results have been reported with a good level of detail. A few questions remain, as the authors noted, relative to the reactivity of the agents towards Lys but also Cys, with kinetics that is apparently very slow. One question to be addressed is the aqueous and plasma stability of the agents (the experiments with the lysates would suggest some stability but it would be nice to have those quantifications given that incubation times are very long: overnight or many hours).

We thank the reviewer for their appreciation of the work and valuable comments. Regarding stability of the peptides, we have estimated their stability in two ways: First, we incubated the peptides in buffer and in the presence of cysteine and lysine to estimate both their intrinsic stability as well as their reactivity towards thiols and amines (supplementary figure 3 and below). The results show that the peptides are indeed quite reactive towards thiols, forming both substitution and addition products, but do not spontaneously react with lysine in solution even after prolonged incubations, in agreement with the significant selectivity exhibited by the peptides in lysates and medium. Extended incubation of many days does result in isomerization reactions in some peptides. We have added the following text (p.8):

“We were interested in comparing the intrinsic reactivity and stability of the active peptides with their labeling rates (supplementary figure 3). We therefore tested their stability in buffer and in the presence of lysine and cysteine at room temperature. The peptides showed good stability in buffer – In a timescale of 6 hours in buffer, all peptides were more than >90% intact. Over a timescale of days, peptides **8** and **11** formed a new product with the same mass as the original peptide, possibly due to an internal reaction within the peptide. Peptide **3** remained unmodified even after 3 days. Incubation with lysine did not result in any products other than those observed after prolonged incubation in buffer, indicating the peptides had low intrinsic reactivity towards lysine. The reactivity towards cysteine was much higher – the peptides reacted with cysteine to generate both Michael adducts as well as substitution products in which the thiol peptide is released. Peptides **3**, **8** and **11** were more than >50% reacted within 2.5 hours”.

Supplementary Figure 3. Peptide stability in buffer and in the presence of lysine and cysteine.

A. Peptides **3**, **8** and **11** were incubated with 1 mM N-acetyl cysteine methyl ester (NAC, magenta) for 2.5 hours, 5 mM N-acetyl-lysine methylester (NAL, gold) for 3 days, and buffer only (HEPES 25 mM pH = 7.5, 100 mM NaCl, 10 mM MgCl₂) for 3 days, and the products were analyzed using LC/MS. The products were identified as shown in panel **B**. Product **g** is an isomerization product of peptides **8** and **11**, possibly due to migration of the methacrylate to a different position in the peptide, as it shows the same molecular mass, but different retention times to product **c**.

Furthermore, to estimate if the peptides retain their activity for extended periods in lysates and medium, we have now compared the degrees of labeling observed after 1 hour and after overnight (22 hr) incubation (figure 5a and below). The results indicate significantly greater labeling of 14-3-3 proteins after overnight incubation, indicating that the peptides maintain stability and activity for extended periods. We've added the following line to p.11:

“...the bands for the methacrylate peptides intensify after a long incubation (22h compared to 1h) indicating their stability under these conditions.”

Figure 5. Labeling of 14-3-3 proteins by BODIPY-modified peptides in A549 lysates and in medium.

It also may be also useful to compare the proposed methacrylates to fluorosulfates (the authors cite correctly sulfonyl fluorides which are more reactive – but fluorosulfates are very stable and less reactive). Aryl-fluoro-sulfates suffer from the same slow kinetics but react more readily when properly juxtaposed to a binding site targeted Lys (see for example Baggio et al. *J. Med. Chem.* 2019, 62, 20, 9188–9200). Hence, the main limitation being the relatively slow kinetics of covalent labeling in comparison to cysteine or Lys targeting probes may also be due to the chosen pairs, and this could be discussed in the manuscript.

We thank the reviewer for raising this point and have now amended the discussion as follows (p. 19):

“Similar issues with reaction rates arise with aryl-fluorosulfates^{106–109}, and the kinetics can be improved significantly by improving the positioning of the electrophile relative to the target residue¹⁰⁶”.

There is a need for novel synthetically accessible electrophiles to expand the druggable space available to covalent ligands, hence I feel this current manuscript supports new applications for the design of covalent chemical probes and pharmacological tools.

Thank you.

Reviewer 2:

1. Target specificity should be better characterized. The peptide fluorescence gel (Supplementary Figure 3) shows additional bands, what are those proteins targeted by the covalent peptide?

Indeed, while the peptides bound 14-3-3 with high selectivity in lysates, some off-target bands were observed. To characterize these targets, we prepared biotinylated derivatives of peptides 3 and 8, incubated them with lysates and performed pull down proteomics using streptavidin beads. The experiment revealed that all 14-3-3 proteins are targeted selectively by the peptides, and were the most prominent targets. Some off-targets were identified as well with lower enrichment values, many of which are NAD / NADP – dependent redox enzymes or other enzymes that have phosphate – containing substrates. We have modified the manuscript as follows (p. 13):

To further characterize the selectivity and off targets of the methacrylate peptides, we synthesized biotinylated derivatives of peptides **3** and **8**, incubated A549 lysates with them, enriched the biotinylated proteins using streptavidin beads and used trypsin digestion followed by LC-MS/MS to characterize the bound proteins. All isoforms of 14-3-3 were bound efficiently and are the most prominent targets with few off-targets, confirming that peptides **3** and **8** were selective, pan-14-3-3 reactive probes (Figure 6). Several off-targets were identified, many of which are NAD / NADP dependent enzymes such as aldo-ketoreductases, aldolases and dehydrogenases. These contain a defined binding pocket for the phosphate containing cofactor with nearby lysine residues^{87,88}. Enzymes with phosphate containing substrates, including several glycolytic enzymes, were also prominent off-targets (Supplementary File 2). We speculate that the phosphorylated peptides may compete for these binding sites and form covalent adducts with these proteins. Nevertheless, the fluorescence imaging results indicate that 14-3-3 proteins are targeted very selectively, and that only a small fraction of the off-targets become modified due to lack of more specific sequence recognition.

Figure 6: Characterization of the selectivity of peptides 3 and 8 using pull-down proteomics. A549 lysates were treated with biotinylated derivatives of peptides **3** or **8** (1 μ M, 22 hours 25°C). The biotinylated proteins were enriched using streptavidin beads, digested with trypsin and analyzed using LC-MS/MS. ”

The effect of lysate concentration should be examined? It seems that only one concentration was assessed and the exact concentration was not specified. The target specificity of the peptide covalent inhibitors should be examined at high lysate concentrations to mimic intracellular environments.

Following the reviewer's suggestion, we have now tested the peptide's labeling with increased lysate concentration and have found it does not affect the labeling selectivity. We have added the following text (p.12) and supplementary Fig. 6B (below)

"We also found that increasing concentration of the lysate during incubation does not affect the selectivity of the peptides (supplementary figure 6B)."

B

Direct gel imaging for experiment in which 1 μ M peptide was incubated with A549 lysates at the indicated concentrations for 22 hours at 25°C. The same amount of lysate was loaded in each lane (28.5 μ g). The intense fluorescence at the bottom comes from unbound peptide.

2. The advantage of CovPepDock is not obvious. The authors identified residues for mutation/warhead incorporation based on the distance to the targeted residue. That can be with just PyMol examining the peptide-protein complex structure. More details should be given on CovPepDock results, how many peptide variants were examined in total computationally? What are the non-obvious details revealed through this docking program that eyeballing the complex structure would not give?

We agree with the reviewer that manual inspection would likely work well, especially in the case of peptide-protein complexes. That said, when many peptides are available for derivatization (and several positions are possible within each), automative and quantifying the candidates may still be useful. We have added Supplementary Table 1 showing the interface backbone RMSD for the best scoring models for 40 different peptide positions modified with the electrophile on the background of various Peptide substrates.

Another aspect in which modeling may prove crucial is when selecting electrophilic modification from a wider pool of possibilities. We have shown, for instance that the ethyl group in peptide 3 could be replaced with a phenyl in peptide 3a. Using CovPepDock we were able to model this substitution (Supplementary Figure 12 B (and below)). In the future we envision expanding the modeling to a wide range of electrophiles and there, manual inspection would not be able to replace modeling.

Finally, for protein covalent reagents, manual inspection and intuition might be less accurate. We added Supplementary Figure 13 (also below) which shows for the Im9/E9 both an example where one might predict a position to be a good place for introduction of the electrophile but no good models were sampled, as well as an example for a position that is not obvious by distance but several good scoring low RMSD models were sampled.

We've also added the following text to the discussion p. 20 to reflect the above:

“Another aspect that contributes to the practicality of our approach is the computational modeling support. While manual inspection and selection of positions for introduction of the electrophile, may work well for a few peptides, automatic modeling and selection can cover larger number of possibilities (Supplementary Table 1). Moreover, as we expand the number of available electrophiles, with variable side-chains, modeling will be required to address the combinatorics (Supplementary Figure 12). Finally, for *protein* covalent reagents (compared to peptides), manual inspection can be more challenging (Supplementary Figure 13).”

Supplementary Figure 12. A) Labeling kinetics of 14-3-3 σ (2 μ M) by peptide **3** and phenyl ester derivative peptide **3a** (25°C, 5 μ M peptide). B) CovPepDock model of peptide **3a** bound to 14-3-3 σ (blue) overlaid on the structure of parent noncovalent peptide (teal).

Supplementary Figure 13. Structural insights provided by CovPepDock for covalent protein reagents. **A.** Top-scoring model of the E9-Im9 complex when mutating Val34 of Im9 to the methacrylate warhead. While this position is located in close proximity to the target Lys97 ($C\alpha$ - $C\alpha$ distance = 7.5Å), none of the 10 top-scoring models had constraint score < 2, indicating a non-ideal covalent bond geometry. **B.** Docking model (RMSD = 0.758Å) of the E9-Im9 complex when mutating Glu42. While this position seems to be pointing away from the target Lys97, CovPepDock revealed a potential shift of the helix that enables the formation of the covalent bond.

Reviewer 3:

Questions related to text:

1, It may be a very naive question from a chemistry point of view, but can the ethyl 2-(bromomethyl)acrylate react with Lys residues of a peptide, or it will always preferably react with Cys? Will it react with a Cys-free peptide that contains a single Lys? How much weaker is the reactivity of the acrylate group to Cys? Is it possible that a side reaction could occur where the bromomethyl remains intact and the peptide is labeled with acrylate? I think for non-chemists, these discussions could come useful. The authors propose a versatile approach that is not limited to use by chemists, but the same community mostly will be unaware of the practical considerations of the approach.

The bromo-methacrylate reagent may indeed react with amines if given sufficient time and high concentrations. We have observed this while trying to prepare the covalent Im9 protein, for which multiple methacrylate-labeled protein was observed if a large excess were added (p.14): “Preparation of methacrylate-modified Im9 mutant under native buffer conditions was impractical as modification of the cysteine was slow and was competed by modification of other sites, as observed by the appearance of multiply-labeled species before full formation of the mono-labeled protein was observed, possibly indicating the cysteine was not fully exposed. Preparation under denaturing conditions (50% acetonitrile) was far more efficient, with rapid and selective modification in the time scale of minutes up to an hour”.

However, cysteines appear to be far more reactive, so if the pH is maintained around 7-7.5, reaction times are not overextended and large excesses are avoided, then the cysteine can be modified very selectively and we get close to homogenously labeled protein. That being said, it is certainly recommended to avoid the presence of amine-containing buffers or thiols in the labeling reaction. We have now added the following text to clarify this in the discussion (p. 20):

“First, the modification of the peptide or protein into the methacrylate should be performed in the absence of reducing agents such as DTT or betamercaptoethanol, and preferably in the absence of high concentrations of amines such as tris buffer. Once the methacrylate peptide is synthesized, it is not prone to non-specific reaction with amines, but remains sensitive to thiols. While this could limit the use of these reagents in reducing environments such as cells, our results show the peptides retain activity in lysates even for prolonged incubations (Figure 5). Furthermore, these reagents may be applied effectively in non-reducing environments such as extracellular media.”

2, Although the authors avoided using primary amine-containing buffers, such as the most commonly used TRIS, but they do not mention it in their work. I think it could be useful to mention if this reaction could work, or not in TRIS buffer. Also, the authors use bME during their purification steps. Is residual bME contamination a concern?

We in fact did not use betamercaptoethanol during purification of the proteins, so the presence of thiols during the modification with 2-(bromomethyl)acrylate was not an issue. Indeed, Tris would

not be recommended for the labeling reaction, because even though cysteines react more readily than amines, millimolar concentrations of tris could definitely compete. Regarding reducing agents, they should definitely be avoided (similarly to labeling with maleimides) and as we have shown the methacrylate-modified peptide is itself sensitive to them (but not after attachment to the protein). To make this clearer to the reader we mention this now in the discussion (p. 20, same quote used previously).

3, What ensures that the warhead will not intramolecularly? Only one of the tested peptides contained an intramolecular Lys residue (peptide 7) and this peptide did not show a reaction with 14-3-3 sigma. What is the author's opinion point regarding this issue? How likely that the warhead could staple the peptide intramolecularly turning a good potential binder inactive? What happens if the peptide has a free N-terminus?

In principle intramolecular reaction of the warhead is possible, as well as a reaction with a free N-terminus (and this reaction would not lead to a mass change). Two lines of evidence show this is not a significant issue – first, none of the peptides react with free lysine in solution even at extended incubation times (days) and high concentrations of lysine (5 mM – supplementary figure 3). Therefore, it appears the intrinsic reactivity of the methacrylate group towards amines is too low to drive the reaction unless optimal positioning is achieved through a previous noncovalent binding. Furthermore, this does not appear to have occurred in the immunity protein – the methacrylate-modified protein was fully capable of forming a covalent adduct with E9, despite having multiple free lysines and a free N terminus, which would not have been possible if it had reacted internally. We would also point out that we observe slow kinetics of the reaction even in cases where the noncovalent binding is potent and positions the lysine in close proximity to the methacrylate.

That being said, peptides 8 and 11 did show an internal reaction after extended (3 days) incubation, but this happens sufficiently slowly to make the peptides stable enough to be effective probes for 14-3-3. We now discuss this point on p. 20: “A second issue is the possibility of an internal reaction between the introduced methacrylate group and a nucleophile such a lysine residue on the peptide or protein, which would inactivate it. However, the peptides did not react with high concentrations of lysine even after extended incubation, and the Im9 methacrylate remained fully capable of reacting with E9 despite the presence of surface lysines on the protein. Peptide **8** and **11** did react internally on the time scale of days, which is slow compared to the reaction with 14-3-3 proteins and did not interfere with labeling in lysates and in media (Figure 5).”

4, Naively looking, peptides with modification at the P+3 position should target this Lys, but none of them reacted with 14-3-3s significantly. Lys49 is involved in phosphate binding. Is it realistic to expect a reaction to this site?

Indeed, no peptides reacted with Lys49. Lys49 was also expected to have a higher pKa and lower reactivity towards electrophiles, which may explain the lack of reaction with it (Wolter et al., *Angewandte Chemie*, 2020, 59, 21520-21524). We have added this point to the text (p. 18): “None of the peptides reacted with Lysine 49, which may be attributed to the lower predicted nucleophilicity of this residue⁸³.”

5, The authors show that the reaction forms relatively slowly, but I have found some discrepancies in their data. In their first screen with their peptide panel, they performed the experiment with 100-fold molar excess overnight (approximately 14-16 h) resulting 30-60% reaction. Then, they perform an experiment with 2.5-fold molar excess for 5 h reaching >80% reaction. By decreasing the concentration of the peptide, I would expect a slower reaction. I think the key behind the observed reaction acceleration is that this reaction is very temperature sensitive and while the first reaction was performed in cold, the second was performed at room temperature. If this is true, how fast could be this reaction at more physiological temperatures (37 degrees)? Given the differences between the reaction rates at different temperatures, the authors should specify in the main text the conditions at each reaction and not only in the method section.

Indeed, the difference is due to temperature differences. We have clarified the temperatures in the manuscript. The initial screen was performed at 4°C to facilitate the testing of multiple samples. After this, time-dependent experiments were conducted at 25°C. Based on the reviewer's suggestion we have now also tested the labeling rates by peptides 3 and 8 at higher temperatures (supplementary figure 2 and below) and found further increase in the labeling rate. These results are discussed in page 7: "We analyzed these peptides further using time course labeling experiments at lower peptide concentrations (5 μM) at 25°C. Peptides 3 and 8 reached 60% and 80% labeling within 5 hours, respectively. Incubation at 37°C increased the rate of labeling roughly 4-fold (supplementary figure 2)."

Supplementary Figure 2. Peptide labeling at various temperatures.

Peptides 3 and 8 (5 μM) were incubated with 14-3-3σ (2 μM) at either 25°C or 37°C and the degree of labeling was measured using LC/MS.

6, For such a slow reaction, it could be possible to measure the dissociation constants of the peptide inhibitor, as well as the reaction rate constant. I think it could be interesting to discuss how labeling the peptide could change its biophysical properties and for example, decipher why only 3-4 peptides out of 11 showed a reaction with 14-3-3 sigma. Is it possible that the other peptides only bind with very weak affinities after labeling?

We decided to approach this question by performing a competition experiment between a previously characterized fluorescent binder for 14-3-3 sigma (with an affinity of ~100 nM) and the screened peptides. Indeed, most of the peptides could not displace the fluorescent binder at 5 μ M, indicating the modifications do affect the affinity. However, at 200 μ M (at which the initial screen was performed), all of the peptides except peptide 2 showed efficient displacement. Therefore, it is likely that in the screening conditions, the protein is mostly bound noncovalently and that the orientation of the electrophile in the complex plays a key role. This experiment was added to the manuscript in supplementary figure 1 (and shown below) and discussed in page 7: “At this point it was not clear whether the peptides that did not label 14-3-3 σ failed to do so due to diminished noncovalent binding affinity or due to suboptimal positioning of the electrophile within the noncovalent complex. Therefore, we exploited the fact that the formation of the covalent bond is slow and takes place over a time scale of hours, and performed a fluorescence polarization binding experiment using a BDP-labeled peptide derived from YAP-1 that binds 14-3-3 σ with an affinity of ~100 nM. We added 14-3-3 σ at a concentration of 0.25 μ M to premixed fluorescent peptide (5 nM) and electrophilic peptides (5 μ M or 200 μ M) and measured the fluorescence polarization at 27°C immediately following the mixture (supplementary figure 1). Only peptides 10 and 11 had sufficient affinity to displace the fluorescent binder at 5 μ M, while all peptides except peptide 2 displaced the binder at 200 μ M, which is the concentration at which the initial screen was performed. Therefore, while many of the electrophilic peptides have diminished noncovalent affinity to 14-3-3 σ , it is likely that the positioning of the electrophile in the noncovalent complex also plays an important role in covalent bond formation.”

Probe peptide: Bodipy-PropargylGly-RAH

SPASL-Dab(acetyl)-NH₂

Supplementary Figure 1, Fluorescence Polarization competition experiments.

Fluorescence polarization of 10 nM BODIPY-labeled noncovalent binder for 14-3-3 σ was measured alone (green), in the presence of 0.25 μ M 14-3-3 σ (blue) and in the presence of 14-3-3 σ and the different electrophilic peptides (orange and magenta). The sequence of the BDP labeled binder is shown on top, Dab = diaminobutyric acid. The control peptide is identical to the BDP-labeled binder in sequence.

7, The authors mention that the adduct is unstable in the presence of reducing agents. This is concerning because most proteins require reducing agents in vitro and because the cellular environment is reducing. What is the expected half-life of this bond, can it be suitable as is in live cell assays?

Indeed, the peptides are not stable to reducing agents at millimolar concentrations, and react with cysteine within about 2 hours via either elimination or addition. Activity in cells is also made difficult by the limited cell permeability of such peptides. Nevertheless, they are still capable of reacting potently and selectively with 14-3-3 proteins in lysates that contain many thiols, both within proteins and as small molecules such as glutathione. Therefore, combined with effective noncovalent recognition, we believe that molecules harboring these electrophiles can function as effective probes and inhibitors. Furthermore, applications such as detecting proteins in serum are not limited by cellular stability, and therefore we believe such peptides have many potential applications despite their sensitivity to reducing agents.

8, The Rfree of the structure with peptide 8 seems to be high compared to the resolution, as well as the gap between Rwork and Rfree is high. I would be curious if the authors could comment on what limited their refinement in this instance.

We thank reviewer 3 for his critical view on our crystal structure and we agree that the R-factor gap is a bit larger than we would ideally like to see. We reason that this gap is most probably caused by some undefined electron density in the Fo-Fc map (especially around the C-terminal site of the peptide and some unstructured loop regions of 14-3-3). These electron densities indicated there are multiple conformation of the peptide/protein, but it was rather challenging to build in one (or two) defined model(s) in the electron density. Therefore, we have decided to not built a model in these regions, as we cannot with full certainty determine the structure/conformation. These undefined electron density regions therefore most probably cause the R-factor gap. Nevertheless, we are confident that the regions that were defined by the model are accurate. To further strengthen this, we have added additional electron density maps in the supplementary information (Supplementary Figure 5 C,D) and amended the legend of Figure 4.

9, Are there any additional Lys or Cys residues that have excess electron density in the structures?

We did not observe any additional electron density around other lysine or cysteine residues in both crystal structures, indicating that the peptide did not covalently bind to other residues besides Lys122.

10, It is a great idea to create 14-3-3 inhibitors that target the whole family instead of a particular isoform. I am curious why the authors choose sigma as their primary target, which appears to be the most divergent out of the family that also appears to bind with markedly weaker affinity to all targets. Would it be possible that the other peptides would react with higher efficiency with other isoforms, such as gamma, or eta?

While it is possible that other isoform would bind the peptide with different potencies, when we tested the binding of peptides 3 and 8 to other 14-3-3 isoforms (supplementary figure 5) we observed only marginal differences in labeling. An important benefit that comes from using the sigma isoform in this study is that it allows us to directly observe the site specificity of labeling by the peptide, even in the presence of a nearby cysteine. Proteomics pull down experiments and LCMS experiments confirmed that the peptides function as pan 14-3-3 binders.

11, I tend to agree with the authors that the BODIPY-modified peptides preferably target 14-3-3s in cell extracts (or in the medium). However, I have some problems with the way how the supporting data is presented. First, specificity cannot be decided based on the cropped images shown in the main figure, because a highly specific off-target can run at a different MW. It would be better to show the entire lanes. Second, the markers need to be labeled better. Either units or complete annotations are missing. In my opinion, multiple bands are labeled with all peptides based on the fluorescence data that shows that neither of these peptides are fully specific. I do not think this is a problem, a full range of other molecules are expected to interact with such molecules such as phosphatases, kinases, or others, but the authors should be more straight with their interpretation.

We performed several repetitions of the western blot experiments and now present the full lanes of the membrane to enable observation of the selectivity. In addition, we also characterized the labeled 14-3-3 proteins as well as the off-targets using proteomics pull-down experiments, as described in page 13: “To further characterize the selectivity and off targets of the methacrylate peptides, we synthesized biotinylated derivatives of peptides **3** and **8**, incubated A549 lysates with them, enriched the biotinylated proteins using streptavidin beads and used trypsin digestion followed by LC-MSMS to characterize the bound proteins. All isoforms of 14-3-3 were bound efficiently with few off-targets, confirming that peptides **3** and **8** were selective, pan-14-3-3 reactive probes (Figure 6). Several off-targets were identified, many of which are NAD / NADP dependent enzymes such as aldo-ketoreductases, aldolases and dehydrogenases. These contain a defined binding pocket for the phosphate containing cofactor with nearby lysine residues. Enzymes with phosphate containing substrates, including several glycolytic enzymes, were also prominent off-targets (Supplementary File 2). We speculate that the phosphorylated peptides may compete for these binding sites and form covalent adducts with these proteins. Nevertheless, the fluorescence imaging results indicate that 14-3-3 proteins are targeted very selectively, and that only a small fraction of the off-targets become modified due to lack of more specific sequence recognition.”

12, Since the 14-3-3 beta antibody stains multiple bands, I wonder if we can consider this as a specific staining. Given the close homology between the 14-3-3 isoforms, I would not be surprised if we would see a pan 14-3-3 pool in the blot. Also, how many times the experiment was repeated?

It is certainly possible the antibody labels multiple isoforms, but we believe this does not affect the validity of the results. Western blot experiments were performed three times.

13, The fact that denaturing of Im9 was essential for labeling is very unfortunate if the authors would like to demonstrate that the labeling approach is compatible with native conditions. Based on the structures, E41 appears to be solvent exposed. Is it possible that the protein forms some intrinsic oligomers that hide this residue? If yes, is it possible that the labeling prevents the formation of the same oligomeric state? I would be curious to see SEC data on these proteins, or CD spectra before and after labeling that could show that neither the oligomeric state Im9, nor the overall conformation of the protein is affected by the chemical labeling.

We thank the reviewer for this suggestion. We performed SEC-MALS experiments to estimate the oligomeric state of the constructs. The data indicate that mutation of Im9 and modification to the methacrylate do not affect the oligomeric state and Im9 remains monomeric in all cases. It appears the mutation does make the protein adopt a slightly more compact conformation that becomes less compact upon modification with the methacrylate. The same is observed for the Im9-E9 complexes. The results therefore indicate that the modification does not significantly alter the structure of the immunity protein or complex. The results are described in page 16: “Although the native binding affinity of these two proteins is very high, we wanted to test whether the mutation and covalent binding affects the structure and stability of the complex. To this end, we analyzed the pure Im proteins and the complexes with E9 using SEC-MALS (supplementary figure 10). The estimated M_w of the proteins from the elution volume agree with the MALS measurements and indicated that Im proteins are monomeric, and that the C23A/E41C mutation in the Im protein makes the protein adopt a slightly more compact conformation, which may explain the difficulty of modifying the protein in native buffers. The methacrylate-modified mutant behaves more similarly to the WT, and the same trends are observed for the complexes with E9. These results indicate that the covalent complex adopts a similar structure to the native complex.”

14, Regarding the thermal stability experiments, I do not find the results particularly interesting from the method point of view and the results are not very surprising. Complexes tend to be more stable than their parts and a covalent complex is expected to be more stable than a transient one. In my opinion, instead of the stability assay, it would have been more interesting to show that the overall conformation of the transient and covalent complex is practically the same with CD spectroscopy or SAXS, or other assays. I would propose moving Figure 7 to the supplement.

As mentioned before we used SEC-MALS to characterize the complexes, and moved the mentioned figure to the supplementary material.

15, The authors should check subscripts and superscripts in their chemical formulas. (E.g. Ni^{2+} , $MgCl_2$, $CaCl_2$...)

We apologize for this and have fixed these errors in the manuscript.

Questions related to figures:

Figure 2D - I get the idea that it is a schematic figure representing the major steps, but I find the lack of sequence disturbing. Is there only a single Cys in the peptide? Are there Lys residues? If you would like to show real data, please include the sequence, or only show schematic data.

We modified the figure to contain the sequence and data for the synthesis of peptide 3.

Figure 3 - I find it difficult to interpret the intermediate complex of peptide 11. Was free ethyl 2-(bromomethyl)acrylate still present at this stage of the experiment? Was not the peptide HPLC purified after labeling? Or the linkage between the warhead and the peptide is fragile and can be broken resulting in a free acrylate compound that could reversibly react with the protein?

The peptide was indeed purified and homogenous. To better understand the reaction dynamics and to assess the sensitivity of the peptides to reducing conditions, we incubated the peptides with 1 mM of N-acetylcysteine methyl ester and characterized the products. The reaction resulted in simultaneous formation of methacrylate-free peptide and the cysteine adduct to the peptide, indicating that substitution and addition are separate reaction mechanisms. We also observed the cysteine-methacrylate (product of the substitution) which over time reacted further with the cysteine (which was in excess) to generate an addition product of two cysteine molecules on the methacrylate. We conclude that when the peptide is incubated with the protein, some of the peptide reacts via addition while some forms a substitution product. Since no other thiols are present in the sample, the methacrylate-modified protein eventually reacts with the liberated thiol peptide via addition and eventually only an addition product is formed. This apparently occurs only when the peptide reacts with a cysteine – reaction with lysine is purely via addition mechanism. We expanded the discussion of the subject in page 17: “The reaction of **11** with the cysteine appears to occur via two distinct mechanisms – addition and substitution, which were also observed when the peptide is incubated with cysteine. In addition, the cysteine adds via Michael addition to the methacrylate, while in substitution, the cysteine displaces the peptide, which is released as a free thiol, while the methacrylate remains on the protein. The methacrylate-labeled protein can later react via addition with free thiol peptide, eventually converting all the protein to an addition product, which is stable (Figure 3B).”

Figure 3 - Peptide 1 shows two bands that are not commented on in the text. Are these meaningful products, too?

Indeed, peptide 1 labeled 14-3-3 with two observed adduct masses, one at 864 Da (the expected adduct) and another, unidentified adduct mass of 725 Da. Due to the low efficiency of labeling by this peptide compared to peptides 3, 8 and 11, we decided not to characterize it further. We updated the manuscript to explain this result. We have referred to it in page 7: “Peptide **1** also displayed low levels of labeling (~10%) at the expected adduct mass, as well an unidentified smaller adduct (139 Da less).”

Figure 4 - What is "electron density"? Is that a final 2FoFc map at a certain sigma level? Or an omit map? The authors need to be more precise on this point. In case this is a final map, the authors should show a bias-free omit map (e.g. simulated annealing omit map) in the supplement.

We apologize to the reviewer for any confusion. This indeed is the final 2Fo–Fc electron density map contoured at 1.0σ . To further clarify this aspect and to be more accurate we have amended the legend of figure 4, describing it is the 2Fo-Fc density map contoured at 1σ . We have added a bias-free omit map to the supplementary information.

Figure 4 - Why Lys49 is shown in the figure?

This was done in order to illustrate it was not modified.

Figure 6C- I get that the authors show a chromatogram, but what sort of chromatogram? Gel filtration, reverse phase HPLC, or something else?

This is a reverse phase HPLC chromatogram. We have clarified this in the manuscript.

SF2C - What are 3, 8, 12? Peptide IDs, molecular weights, time, or temperature? Clarify labels better in general, deciphering figures should be less guesswork.

We have clarified the labeling in the supplementary figures.

REVIEWERS' COMMENTS

Reviewer #1 (Remarks to the Author):

The authors have clarified my observations with two new detailed experiments and associated figures (one reported as supplementary and one new) and respective clarifying statements in the manuscript. These data reinforce my original opinion that the manuscript presents novel and exciting findings.

Reviewer #2 (Remarks to the Author):

The reviewer has done a good job revising the manuscript. All my concerns raised before are properly addressed. Hence, I am happy to support publication of this work in Nature Communications.

Reviewer #3 (Remarks to the Author):

I really appreciate all the modifications the authors did to their manuscript and their responses to my comments. I fully recommend their revised work for acceptance and publication.